# Training Certifiably Robust Neural Networks with Efficient Local Lipschitz Bounds

**Yujia Huang**[1]  **Huan Zhang**[2]  **Yuanyuan Shi**[3]  **J. Zico Kolter**[2,4]  **Anima Anandkumar**[1,5]
[1]California Institute of Technology  [2]Carnegie Mellon University  [3]UC San Diego
[4]Bosch Center for AI  [5]NVIDIA
`yjhuang@caltech.edu huan@huan-zhang.com yyshi@eng.ucsd.edu`
`zkolter@cs.cmu.edu anima@caltech.edu`

## Abstract

Certified robustness is a desirable property for deep neural networks in safety-critical applications, and popular training algorithms can certify robustness of a neural network by computing a global bound on its Lipschitz constant. However, such a bound is often loose: it tends to over-regularize the neural network and degrade its natural accuracy. A tighter Lipschitz bound may provide a better tradeoff between natural and certified accuracy, but is generally hard to compute exactly due to non-convexity of the network. In this work, we propose an efficient and trainable *local* Lipschitz upper bound by considering the interactions between activation functions (e.g. ReLU) and weight matrices. Specifically, when computing the induced norm of a weight matrix, we eliminate the corresponding rows and columns where the activation function is guaranteed to be a constant in the neighborhood of each given data point, which provides a provably tighter bound than the global Lipschitz constant of the neural network. Our method can be used as a plug-in module to tighten the Lipschitz bound in many certifiable training algorithms. Furthermore, we propose to clip activation functions (e.g., ReLU and MaxMin) with a learnable upper threshold and a sparsity loss to assist the network to achieve an even tighter local Lipschitz bound. Experimentally, we show that our method consistently outperforms state-of-the-art methods in both clean and certified accuracy on MNIST, CIFAR-10 and TinyImageNet datasets with various network architectures.

## 1 Introduction

With the ever-growing deployment of deep neural networks, formal robustness guarantees are needed in many safety-critical applications. Strategies to improve robustness such as adversarial training only provide empirical robustness, without formal guarantees, and many existing adversarial defenses have been successfully broken using stronger attacks [1]. In contrast, certified defenses give formal robustness guarantees that any norm-bounded adversary cannot alter the prediction of a given network.

Bounding the global Lipschitz constant of a neural network is a computationally efficient and scalable approach to provide certifiable robustness guarantees [2–4]. The global Lipschitz bound is typically computed as the product of the spectral norm of each layer. However, this bound can be quite loose because it needs to hold for *all* points from the input domain, including those inputs that are far away from each other. Training a network while constraining this loose bound often imposes to high a degree of regularization and reduces network capacity. It leads to considerably lower clean accuracy in certified training compared to standard and adversarial training [5, 6].

A local Lipschitz constant, on the other hand, bounds the norm of output perturbation only for inputs from a small region, usually selected as a neighborhood around each data point. It produces a

35th Conference on Neural Information Processing Systems (NeurIPS 2021).

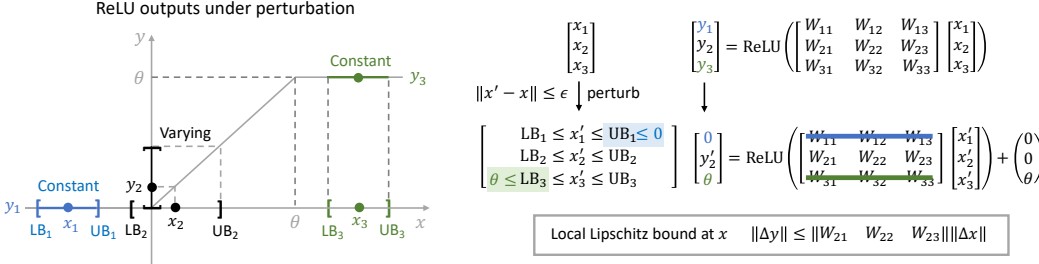

Figure 1: Illustration of tighter (local) Lipschitz constant with bounded ReLU.

tighter bound by considering the geometry in a local region and often yields much better robustness certification [7, 8]. Unfortunately, computing the exact local Lipschitz constant is NP-complete [9]. Obtaining reasonably tight local Lipschitz bounds via semidefinite programming [10] or mixed integer programming [11] is typically only applicable to small, *previously trained networks* since it is difficult to parallelize the optimization solver and make it differentiable for training. On the other hand, many existing certified defense methods have achieved success by using a training-based approach with a relatively weak but efficient bound [12–14]. Therefore, to incorporate local Lipschitz bound in training, a computationally efficient and training-friendly method must be developed.

**Our contributions:** We propose an efficient method to incorporate a local Lipschitz bound in training deep networks, by considering the interactions between an activation layer such as a Rectified Linear Unit (ReLU) layer and a linear (or convolution) layer. Our bound is computed for each data point and this translates to different types of outputs from the activation function: constant or varying under input perturbations. If the outputs of some activation neurons are constant under local perturbation, we eliminate the corresponding rows in the previous weight layer and the corresponding columns in the next weight layer, and then compute the spectral norm of the reduced matrix.

Our main insight is to use training to make the proposed local Lipschitz bound tight. This is different from existing works that find local Lipschitz bound for a fixed network [8, 10, 11]. Instead, we aim to enable a network to learn to tighten our proposed local Lipschitz bound during training. To achieve this, we propose to clip the activation function with an individually learnable threshold $\theta$. Take ReLU for example, the output of a "clipped" ReLU becomes a constant when input is greater than this threshold (see Figure 1). Once the input of the ReLU is greater than the threshold or less than 0, then this ReLU neuron does not contribute to the local Lipschitz constant, and thus the corresponding row or column of weight matrices can be removed. We also apply this method to non-ReLU activation functions such as MaxMin [15] to create constant output regions. Additionally, we also use a hinge loss function to encourage more neurons to have constant outputs. Our method can be used as a plug-in module in existing certifiable training algorithms that involve computing Lipschitz bound. Our contributions can be summarized as:

- To the best of our knowledge, we are the first to incorporate a local Lipschitz bound during training for certified robustness. Our bound is provably tighter than the global Lipschitz bound and is also computationally efficient for training.

- We propose to use activation functions with learnable threshold to encourage more fixed neurons during training, which assists the network to learn to tighten our bound. We show that more than 45% rows and columns can be removed from weight matrices.

- We consistently outperform state-of-the-art Lipschitz based certified defense methods for $\ell_2$ norm robustness. On CIFAR-10 with perturbation $\epsilon = \frac{36}{255}$, we obtain 54.3% verified accuracy with ReLU activation function and 60.7% accuracy with MaxMin [15], outperforming the SOTA baselines, and also achieve better clean accuracy. Our code is available at https://github.com/yjhuangcd/local-lipschitz.

## 2  Related work

**Bounds on Local Lipschitz Constant** A *sound upper bound* of local Lipschitz constant (simply referred to as "local Lispschitz bound" in our paper) is a crucial property to determine the robustness

of a classifier. Finding an exact local Lipschitz constant for a neural network is generally NP hard [16], so most works focus on finding a sound upper bound. Hein and Andriushchenko [7] derived an analytical bound for 2 layer neural networks and found that local Lipschitz bounds could be much tighter than the global one and give better robustness certificates. RecurJac [8] is a recursive algorithm that analyzes the local Lipschitz constant in a neural network using a bound propagation [14] based approach. FastLip [17] is a special and weaker form of RecurJac. Fazlyab et al. [10] used a stronger semidefinite relaxation to compute a tighter bound of local Lipschitz constant. Jordan and Dimakis [11] formulated the computation of Lipschitz as an mixed integer linear programming (MILP) problem and they were able to solve the exact local Lipschitz constant. Although these approaches can obtain reasonably tight and sound local Lipschitz constants, none of them have been demonstrated effective for training a certifiably robust network, where high efficiency and scalability are required. Note that although an empirical estimate of local Lipschitz constant can be easily found via gradient ascent (e.g., the local lower bounds reported in [4]), it is not a sound bound and does not provide certifiable robustness guarantees.

**Certifiably Robust Training using Lipschitz constants**   The Lipschitz constant plays a central role in many works on training a certifiably robust neural network, especially for $\ell_2$ norm robustness. Since naturally trained networks usually have very large global Lipschitz constant bounds [18], most existing works train the network to encourage small a Lipschitz bound. Cisse et al. [2] designed networks with orthogonal weights, whose Lipschitz constants are exactly 1. As this can be too restrictive, later works mostly use power iteration to obtain per-layer induced norms, whose product is a Lipschitz constant. Lipschitz Margin Training (LMT) [19] and Globally-Robust Neural Networks (Gloro) [4] both upper bound the worst margin via global Lipschitz constant with different loss functions. LMT constructs a new logit by adding the worst margin to all its entries except the ground truth class. Gloro construct a new logit with one more class than the original logit vector, determines whether the input sample can be certified. However, these approaches did not exploit the available local information to tighten Lipschitz bound and improve certified robustness. Box constrained propagation (BCP) [20] achieves a tighter outer bound than global Lipschitz based outer bound, by taking local information into consideration via interval bound (box) propagation. They compute the worst case logit based on the intersection of a (global) ball and a (local) box. Although box propagation considers local information, the ball propagation still uses global Lipschitz constant, and its improvement is still limited with low clean accuracy.

**Other Certified Defenses**   Besides using Lipschitz constants, one of the most popular certifiable defense against $\ell_\infty$ norm bounded inputs is via the convex outer adversarial polytope [21, 22]. [13] takes a similar approach via abstract interpretation. These methods uses linear relaxations of neural networks to compute an outer bound at the final layer. However, because the convex relaxations employed are relatively expensive, these methods are typically slow to train. A simple and fast certifiable defense for $\ell_\infty$ norm bounded inputs is interval bound propagation (IBP) [12, 13]. Since the IBP bound can be quite loose for general networks, its good performance relies on appropriate hyper-parameters. CROWN-IBP [14] outperforms previous methods by combining IBP bound in a forward bounding pass and a tighter linear relaxation bound in a backward bound pass. [23] improved IBP with better initialization to accelerate training. Additionally, randomized smoothing [24–26] is a probabilistic method to certify $\ell_2$ norm robustness with arbitrarily high confidence. The prediction of a randomized smooth classifier is the most likely prediction returned by the base classifier that is fed by samples from a Gaussian distribution. Salman et al. [27] further improves the performance of randomized smoothing via adversarial training.

## 3   Method

We begin with notations and background for Lipschitz bound. We introduce our method for local Lipschitz bound computation in Section 3.2. Then we introduce how to incorporate our efficient local Lipschitz bound in robust training (Section 3.3).

## 3.1 Notation and Background

**Notations.** We denote the Euclidean norm of a vector $x$ as $\|x\|$ and $\|A\|$ is the spectral norm of matrix $A$. Subscript of vector $x$ denotes element, i.e., $x_i$ is the i-th element of $x$. We use $\text{LB}^l$ and $\text{UB}^l$ to denote lower bounds and upper bounds of pre-ReLU activation values for layer $l$.

**Definition 1.** *The Lipschitz constant of a function $f : \mathbb{R}^d \to \mathbb{R}^m$ over an open set $\mathcal{X}$ is defined as,*

$$L(f, \mathcal{X}) := \sup_{x,y \in \mathcal{X}, x \neq y} \frac{\|f(y) - f(x)\|}{\|x - y\|},$$

*If $L(f, \mathcal{X})$ exists and is finite, we say that $f$ is Lipschitz continuous over $\mathcal{X}$. Suppose $\mathcal{X} = dom(f)$, $L(f, \mathcal{X})$ is the **global** Lipschitz constant of $f$; if $\mathcal{X}$ is defined as the $\epsilon$-ball at point $x$, i.e., $\mathcal{X} := \{x' | \|x - x'\| \leq \epsilon\}$, then $L(f, \mathcal{X})$ is the **local** Lipschitz constant of $f$ at $x$.*

**Global Lipschitz bound in existing works** Consider a $L$-layer ReLU neural network which maps input $x$ to output $z^{L+1} = F(x; W)$ using the following architecture, for $l = 1, ..., L - 1$

$$z^1 = x; \quad z^{l+1} = \phi(W^l z^l); \quad z^{L+1} = W^L z^L \tag{1}$$

where $W = \{W^{1:L}\}$ are the parameters, and $\phi(\cdot) = \max(\cdot, 0)$ is the element-wise ReLU activation functions. Here we consider the bias parameters to be zero because they do not contribute to the Lipschitz bound. Since the Lipschitz constant of ReLU activation $\phi(\cdot)$ is equal to 1, a global Lipschitz bound of $F$ is,

$$L_{\text{glob}} \leq \|W^L\| \cdot \|W^{L-1}\| \cdots \|W^1\| \tag{2}$$

where $\|W^l\|$ equals the spectral norm (maximum singular value) of the weight matrix $W^l$. However, the global Lipschitz bound ignores the highly nonlinear property of deep neural networks. In what follows, we introduce our method that considers the interaction between ReLU and linear layer to obtain a tighter local Lipchitz bound in a computationally efficient way, that allows us to train a certifiably robust network using local Lipschitz bounds.

## 3.2 Our Approach for Efficient Local Lipschitz Bound

In this section, we use ReLU as an example to describe how we compute our efficient local Lipschitz bound. We will discuss how to apply our method on other types of activation functions in Section 3.3. To exploit the piece-wise linear properties of ReLU neurons, we discuss the outputs of ReLU case by case. Intuitively, if the input of a ReLU neuron is always less or equal to zero, its output will always be zero, which is a constant and not contributing to Lipschitz bound. If the input of a ReLU's can sometimes be greater than zero, the ReLU output will vary based on the input.

We define diagonal indicator matrices $I_{\text{V}}^l(z^l)$ to represent the entries where the ReLU outputs are *varying* and $I_{\text{C}}^l(z^l)$ for entries where the ReLU outputs are *constant* under perturbation. Here $z^l \in \mathbb{R}^{d_l}$ denotes the feature map of input $x$ at layer $l$. Throughout this paper, unless otherwise mentioned, the indicator matrix is a function of the feature value $z^l$, evaluated at a given input $x$.

Given an input perturbation $\|x' - x\| \leq \epsilon$, suppose $z^l(x')$ is bounded element-wise as $\text{LB}^l \leq z^l(x') \leq \text{UB}^l$, we define diagonal matrix $I_{\text{V}}^l$ and $I_{\text{C}}^l$ as:

$$I_{\text{V}}^l(i, i) = \begin{cases} 1 & \text{if } \text{UB}_i^l > 0 \\ 0 & \text{otherwise} \end{cases}, \quad I_{\text{C}}^l(i, i) = \begin{cases} 1 & \text{if } \text{UB}_i^l \leq 0 \\ 0 & \text{otherwise} \end{cases} \tag{3}$$

By this definition, the ReLU output can either be constant or vary with respect to input perturbation. Hence we have $I_{\text{V}}^l + I_{\text{C}}^l = I$, where $I$ is the identity matrix. LB and UB can be obtained cheaply from interval bound propagation [12] or other bound propagation mechanisms [21, 28].

A crucial observation is that to compute the local Lipschitz bound, we only need to consider the ReLU neurons which are non-fixed. The fixed ReLU neurons are always zero (locally, in the prescribed neighborhood around $x$) and thus have no impact to final outcome. We define a diagonal matrix $D_{\text{V}}$ to represent the ReLU outputs that are varying. Then, a neural network function (denoted as $F(x; W)$) can be rewritten as:

$$\begin{aligned} F(x; W) &= W^L D_{\text{V}}^{L-1} W^{L-1} \cdots D_{\text{V}}^1 W^1 x \\ &= (W^L I_{\text{V}}^{L-1}) D_{\text{V}}^{L-1} (I_{\text{V}}^{L-1} W^{L-1} I_{\text{V}}^{L-2}) \cdots D_{\text{V}}^1 (I_{\text{V}}^1 W^1) x \end{aligned} \tag{4}$$

where

$$D_V^l(i,i) = \begin{cases} \mathbb{1}(\text{ReLU}(z_i^l) > 0) & \text{if } I_V^l(i,i) = 1 \\ 0 & \text{if } I_V^l(i,i) = 0 \end{cases},\tag{5}$$

where $\mathbb{1}$ denotes an indicator function.

Note here that we ignore bias terms for simplicity. Based on (4), an important insight used in our approach is that by combining the ReLU function with weight matrix, we have the opportunity to tighten Lipschitz bound by considering $I_V^l W^l I_V^{l-1}$ as a whole based on whether there are ReLU outputs stay at constant under perturbation. Importantly, since $D_V^l$ depends on UB, (4) only holds in a local region of $x$, which leads to a local Lipschitz constant bound at input $x$:

$$L_{\text{local}}(x) \le \|W^L I_V^{L-1}\|\|I_V^{L-1} W^{L-1} I_V^{L-2}\| \cdots \|I_V^1 W^1\|\tag{6}$$

The following theorem states that the local Lipchitz bound calculated via (6) is always tighter than the global Lipchitz bound in Eq (2), for all inputs.

**Theorem 1** (Tighter Lipchitz Bound). *For any input $x \in \mathbb{R}^n$ and L-layer ReLU neural network $F(x; W)$, the local Lipschitz bound calculated via (6) in any neighborhood of $x$ is no larger than the global Lipschitz bound in Eq (2), i.e., $\forall x, L_{local}(x) \le L_{glob}$.*

The proof of Theorem 1 leverages the following proposition.

**Proposition 1.** *If a column and/or row is added to a matrix, then the matrix spectral norm (maximum singular value) will be no less than the spectral norm of the original matrix. That is, given matrix $A \in \mathbb{R}^{m \times n}$, and $y \in R^m, z \in R^n$, then*

$$\sigma_{max}([A|y]) \ge \sigma_{max}(A) \, , \sigma_{max}(\begin{bmatrix} A \\ z \end{bmatrix}) \ge \sigma_{max}(A).\tag{7}$$

*Proof of Proposition 1.* Let $A' = \begin{bmatrix} A \\ z \end{bmatrix}$. The singular value of $A'$ is defined as the square roots of the eigenvalues of $A'^T A'$, where $A'^T A' = A^T A + z^T z \ge A^T A$, that simply imply $||A'|| = \sigma_{max}(A') \ge \sigma_{max}(A) = ||A||$. Similar holds for $A' = [A|y]$. $\square$

By Proposition 1, it is straightforward to show that $\|I_V^{L-1} W^{L-1} I_V^{L-2}\| \le \|W^{L-1}\|$ since the left hand side is the spectral norm of the reduced matrix, after removing corresponding rows/columns in $W^{L-1}$ where the neuron output under local perturbation is constant. Therefore, the product of spectral norm of the reduced matrices is no larger than the product of the spectral norm of raw weight matrices, which leads to $\forall x, L_{local}(x) \le L_{glob}$.

## 3.3 Training for Tight Local Lipschitz

To encourage the network to learn which rows and columns need to be eliminated to make local Lipschitz bound tighter, we combine our local Lipschitz bound computation with certifiably robust training. This is different from existing works leveraging optimization tools to find local Lipschitz bound for a fixed network [10, 11]. By training with the proposed local Lipschitz bound, we can achieve good certified robustness on large neural networks.

More precisely, using our local Lipschitz bound, we can obtain the worst case logit $z^*$ that is used to form a robust loss for training: $\mathbb{E}_{(x,y)\sim\mathcal{D}}\mathcal{L}(z^*(x), y)$, where $\mathcal{L}$ is the cross entropy loss function, $(x, y)$ is the image and label pair from the training datasets. A simple way to compute the worst logit is $z_i^* = z_i + \sqrt{2}\epsilon L_{\text{Local}}$ for $i \ne y$, $z_y^* = z_y$ (see [19]). Our approach is also compatible with tighter bounds on the worst case logit, such as the one used in BCP [20]. To give the network more capability to learn to tighten our proposed local Lipschitz bound, we propose the following approaches:

**Allowing More Eliminated Rows via ReLU$\theta$**   The key to tighten our local Lipschitz bound is to delete rows and columns in weight matrices that align with singular vectors corresponding to the largest singular value. To encourage more rows and columns to be deleted, we need to have more ReLU outputs to be at constant under perturbation. Standard ReLU is only lower bounded by zero, but is not upper bounded. If we can set an "upper bound" of ReLU output, we can have more neurons

have fixed outputs at this upper bound. An upper bounded ReLU unit called ReLU6 is proposed in [29], where the maximum output is set to a constant 6. Different from ReLU6 that sets a constant maximum output threshold, we make the threshold to be a learnable parameter. We name the new type of activation function $\text{ReLU}\theta$, which is defined as:

$$\text{ReLU}\theta \left(z_i; \theta_i\right) = \begin{cases} 0, & \text{if } z_i <= 0 \\ z_i, & \text{if } 0 < z_i < \theta_i \\ \theta_i, & \text{if } z_i >= \theta_i \end{cases} \tag{8}$$

where $\theta_i$ is a learnable upper bound of the ReLU output.

Similar to (3), the indicator matrices for the varying outputs of $\text{ReLU}\theta$ are

$$I_V^l(i, i) = \begin{cases} 1 & \text{if } \text{UB}_i^l > 0 \text{ and } \text{LB}_i^l < \theta_i \\ 0 & \text{otherwise} \end{cases} \tag{9}$$

Depending on the $\text{ReLU}\theta$ activation status, the output of a $\text{ReLU}\theta$ neural network is,

$$F(x; W, \theta) = W^L(D_V^{L-1} W^{L-1} \cdots (D_V^1 W^1 x + D_\theta^1) \cdots + D_\theta^{L-1}), \tag{10}$$

where $D_\theta^l$ denotes the ReLU output fixed at the maximum output value,

$$D_\theta^l(i, i) = \begin{cases} \theta_i & \text{if } I_\theta^l(i, i) = 1 \\ 0 & \text{if } I_\theta^l(i, i) = 0 \end{cases}, \quad I_\theta^l(i, i) = \begin{cases} 1 & \text{if } \text{LB}_i^l \geq \theta_i \\ 0 & \text{otherwise} \end{cases} \tag{11}$$

The local Lipchitz bound is still calculated as (6). However, the bound can be potentially learned tighter because we encourage ReLU outputs to be constant in both directions, and there could be less varying outputs in Eq (9) than in Eq (3).

**Extension to non-ReLU activation functions**    Our local Lipschitz bound can be applied on non-ReLU activation functions. The key is to create constant output regions for the activation function and delete the corresponding rows or columns in the weight matrices. Since the MaxMin activation function [15] has been shown to outperform ReLU on certified robustness [4, 30], we take MaxMin as an example to explain how to apply our local Lipschitz bound. Let $x_1$ and $x_2$ be two groups of the input, the output of MaxMin is $\max(x_1, x_2), \min(x_1, x_2)$. To exploit local Lipschitz, we created a clipped version of MaxMin, similar to $\text{ReLU}\theta$. The output of the clipped MaxMin is $\min(\max(x_1, x_2), a), \max(\min(x_1, x_2), b)$, where $a$ is a learnable upper threshold for the max output in MaxMin and $b$ is a learnable lower threshold for the min output in MaxMin. Box propagation rule through MaxMin is straightforward to derive, so we can get the box bound on each entry after MaxMin. If the lower bounds of the Max entries are bigger than the upper threshold $a$, or the upper bounds of the Min entries are smaller than the lower threshold $b$, we can delete the corresponding columns in the successive matrix (similar to the procedure for ReLU networks).

**Encouraging Fixed Neurons via a Sparsity Loss**    To encourage more rows and columns to be deleted in the weight matrices, We design a sparsity loss to regularize the neural network towards this goal. For a ReLU neural network, assuming that the $i$-th entry of the feature map at layer $l$ is bounded by $\text{LB}_i^l \leq z_i^l \leq \text{UB}_i^l$, we hope the neural network can learn to make as many $\text{UB}_i$ to be smaller than zero and $\text{LB}_i$ to be larger than $\theta_i$ without sacrificing too much of classification accuracy. For a MaxMin neural network, let $\text{LB}_{\max}$ be the lower bounds of the Max entries, and $\text{UB}_{\min}$ be the upper bounds of the Min entries. We hope the neural network can learn to make as many $\text{LB}_{\max}$ to be larger than the upper threshold $a$ and $\text{UB}_{\min}$ to be smaller than the lower threshold $b$. The sparsity losses for ReLU and MaxMin networks are as follows:

$$\mathcal{L}_{\text{sparsity}}^{\text{ReLU}} = \max(0, \text{UB}_i^l) + \max(0, \theta_i - \text{LB}_i^l), \quad \mathcal{L}_{\text{sparsity}}^{\text{MaxMin}} = \max(0, \text{UB}_{\min} - b) + \max(0, a - \text{LB}_{\max}) \tag{12}$$

Finally, our full training procedure is presented in Algorithm 1.

**Computational cost**    To obtain the local Lipschitz bound, we must perform power iterations to compute the spectral norm of reduced weight matrices for every input and its feature maps at each layer. Compared to other methods that use optimization tools (e.g., SDP, MILP) to bound local Lipschitz, our method is computationally efficient since only matrix vector multiplication is used.

**Algorithm 1:** Local Lipchitz based Certifiably Robust Training

---

**Input :** Training data $(x, y) \sim \mathcal{D}$, perturbation size $\epsilon$, number of iterations for power method $n$, a neural network with $L$ layers.

**repeat**

> Compute the box outer bound $[\text{LB}^l, \text{UB}^l]$ for layers $1$ to $L$ ;
> Compute indicator matrix $I_\text{V}$ using Eq (9) ;
> // *Compute local Lipcthiz bound $L(x)$ for every input $x$ using Eq (6)*
> **for** *layer from 1 to $L$* **do**
>> // *Power method*
>> Initialize $u^l$ with the updated $u^l$ from the previous training episode ;
>> **for** *$i < n$* **do**
>>> If layer is conv:
>>> $v \leftarrow I_\text{V}^l \text{conv}(W^l, I_\text{V}^{l-1} u^l) / \|I_\text{V}^l \text{conv}(W^l, I_\text{V}^{l-1} u^l)\|$
>>> $u \leftarrow I_\text{V}^{l-1} \text{conv}^\intercal(W^l, I_\text{V}^l v) / \|I_\text{V}^{l-1} \text{conv}^\intercal(W^l, I_\text{V}^l v)\|$
>>> If layer is linear:
>>> $v \leftarrow I_\text{V}^l W^l I_\text{V}^{l-1} u^l / \|I_\text{V}^l W^l I_\text{V}^{l-1} u^l\|$
>>> $u \leftarrow I_\text{V}^{l-1} W^{l\intercal} I_\text{V}^l v / \|I_\text{V}^{l-1} W^{l\intercal} I_\text{V}^l v\|$
>>
>> **end**
>> If layer is conv: $\sigma^l(x) \leftarrow v^l I_\text{V}^l \text{conv}(W^l, I_\text{V}^{l-1} u^l)$
>> If layer is linear: $\sigma^l(x) \leftarrow v^l I_\text{V}^l W^l I_\text{V}^{l-1} u^l$
>
> **end**
> Compute the worst logits using our local Lipschitz bound ;
> Update model parameters based on some loss functions (e.g., Cross-entropy loss).

**until** *training ends*;

**return** *Parameters of a robust neural network*

---

During training, compared to methods that only uses global Lipschitz bound, the local Lipcshitz bound varies based on the inputs. For global Lipschitz bound, a common practice is to keep track of the iterate vector $u$ in power iteration, and use it to initialize the power iteration in the next training batch. With this initialization, only a few number of iterations is performed during training (typically the number of iterations is between 1 to 10 [20, 4]). To extend the initialization strategy for $u$ to compute local Lipschitz, we need to keep track of the iterate vectors based on every input and its feature maps for every layer. Fortunately, this vector only provides an initialization for power iteration so it does not need to be stored accurately, and can be stored using low precision tensors. Further extensions could use dimension reduction or compression methods to store these vectors, or learn a network along the way to predict a good initializer for power iteration. An alternative approach is to random initialize $u$ in every mini-batch, but we find that more power iterations need to be performed during training for this approach to have comparable performance as the using saved $u$.

During evaluation, to minimize the extra computational cost, we can avoid bound computation for two types of inputs: inputs that can already be certified using global Lipschitz bound or can be attacked by adversaries (e.g., a 100-step PGD attack). In practice, this typically rules out more than $80\%$ samples on CIFAR-10 or larger datasets, and greatly reduce computational cost required to compute local Lipschitz constants. In Section 4 we will show more empirical results on this aspect.

## 4   Experiment

In this section, we first show our method achieves tighter Lipschitz bounds in neural networks. When combined with certifiable training algorithms such as BCP [20] and Gloro [4], as well as training algorithms using orthogonal convolution and MaxMin activation function [30], our method achieves both higher clean and certified accuracy.

**Experiment setup**   We train with our method to certify robustness within a $\ell_2$ ball of radius $1.58$ on MNIST [31] and $36/255$ on CIFAR-10 [32] and Tiny-Imagenet [1] on various network architectures.

---

[1] https://tiny-imagenet.herokuapp.com

We denote neural network architecture by the number of convolutional layers and the number of fully-connected layers. For instance, 6C2F indicates that there are 6 convolution layers and 2 fully-connected layers in the neural network. Networks have ReLU activation function unless mentioned otherwise. For more details and hyper-parameters in training, please refer to Appendix C. Our code is available at https://github.com/yjhuangcd/local-lipschitz.

**Tighter Lipschitz bound** We compared the training process of our method and BCP [20] in Figure 2 (a-c). During training, our method uses local Lipschitz bound while BCP uses global Lipschitz bound for robust loss. We also tracked the global Lipschitz bound during our training and the average local Lipschitz bound (computed by our method) during BCP training for comparison. We can see from Figure 2 (a) that our local Lipschitz bound is always tighter the global Lipschitz bound. Furthermore, it is crucial to incorporate our bound in training to enable the network to learn to tighten the bound. We can see that if we directly apply our method to a BCP trained network after training, the local Lipschitz bound has much less improvement over global Lipschitz bound. In addition, a tighter local Lipschitz bound by our method allows the neural networks to have larger global Lipschitz bound in the beginning of training. This potentially provides larger model capacity and eases the training in the early stage. As a consequence, we see a large improvement of both clean loss (Figure 2 (b)) and also robust loss (Figure 2 (c)) throughout the training.

**Sparsity of varying ReLU outputs** We examine our 6C2F model trained on CIFAR-10 and report the proportion of varying (non-constant) ReLU outputs at all layers except the last fully-connected layer (Figure 3). We compared the proportion with that of a standard CNN (trained with only cross entropy loss on unperturbed inputs) and a robust CNN trained by BCP. As we can see, the standard neural networks has the most varying ReLU neurons, indicating that dense varying ReLU outputs may provide larger model capacity for clean accuracy but reduce robustness. Our method has *more* varying ReLU outputs than BCP, while achieving *tighter* local Lipschitz bound than BCP. This indicates that our training method encourages the neural network to learn to delete rows and columns that contribute most to local Lipschitz constant during training, while keeping ReLUs for other rows or columns varying to obtain better natural accuracy.

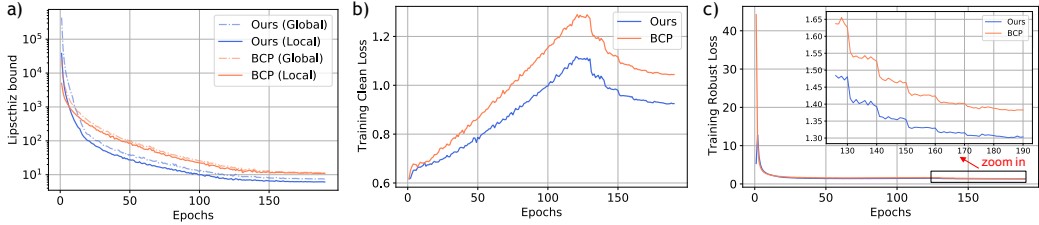

Figure 2: Certifiable training with our method and BCP on CIFAR-10. a) Global and average local Lipschitz bound during training. Cross entropy loss b) on natural and c) on the worst logits.

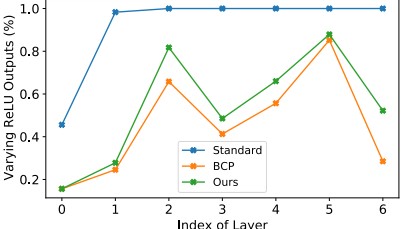

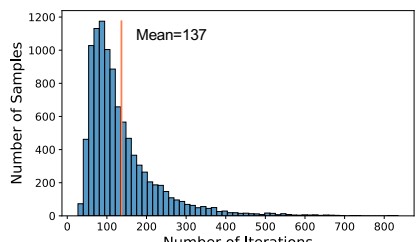

Figure 3: Proportion of ReLU neurons that vary (ReLU outputs are not constants, see definition in Section 3.2) under perturbation.

Figure 4: Histogram for the number of power iterations to ensure convergence for the second last linear layer of the 6C2F CIFAR-10 network.

**Certified robustness** Our method can be used as a plug-in module in certifiable training algorithms that involves using $\ell_2$ Lipschitz constant such as BCP [20] and Gloro [4]. We use Local-Lip-G and

Local-Lip-B to denote our method trained with the Gloro loss and BCP loss respectively (detailed formulation for each loss can be found in Appendix B). We compare the performance of our method against competitive baselines on both ReLU neural networks [20, 4, 19, 12, 22, 33] and MaxMin neural networks [4, 30]. For each method, we report the clean accuracy (accuracy on non-perturbed inputs), the PGD accuracy (accuracy on adversarial inputs generated by PGD attack [6]), and the certified accuracy (the proportion of inputs that can be correctly classified and certified within $\epsilon$-ball). For PGD attack, we use 100 steps with step size of $\epsilon/4$. In our experiments, we use box propagation (as done in BCP) to obtain the lower bound and upper bound of every neuron. With our tighter Lipschitz bound, we further improve clean, PGD, and certified accuracy upon BCP and Gloro, and achieve the state-of-the-art performance on certified robustness (Table 2). On CIFAR-10 with ReLU activation function, we improved certified accuracy from $51.3\%$ (SOTA) to $54.3\%$. When MaxMin activation function is used, our local Lipschitz training approach also consistently improves clean, PGD and verified accuracy over baselines on both CIFAR-10 and TinyImageNet datasets.

To demonstrate the effectiveness of incorporating our bound in training, we also use our local bound to directly compute certified accuracy for *pretrained* models using BCP. Our bound improves the certified accuracy of a pretrained BCP model from $51.3\%$ (reported in Table 2) to $51.8\%$. The improvement is less than training with our bound ($54.3\%$ in Table 2). Therefore, it is crucial to incorporate our bound in training to gain non-trivial robustness improvements.

**Initialization strategy for power method**  We used two initialization strategies for singular vectors $u$ in power method during training. One option is to store $u$ for all the inputs and feature maps and initialize $u$ in the current training epoch with $u$ from the previous epoch, which requires storage. The other option is to random initialize $u$. Table 1 shows the performance of these two approaches on CIFAR-10 with the 6C2F architecture. The number of power iterations during training is listed in the bracket. Although random initialization is memory-efficient, it needs more power iterations during training to achieve comparable performance compared to the approach of storing $u$. Too few iterations tend to cause inaccurate singular value and overfitting, resulting in lower certified accuracy.

| Method | Clean (%) | PGD (%) | Certified (%) |
|---|---|---|---|
| Random init. (2 iters) | 76.7 | 69.0 | 0.5 |
| Random init. (5 iters) | 73.7 | 66.8 | 46.0 |
| Random init. (10 iters) | 72.0 | 65.8 | 51.6 |
| Using saved $u$ (2 iters) | 70.7 | 64.8 | **54.3** |

Table 1: Influence of initialization strategy used in power method. All numbers are accuracy of the 6C2F architecture on CIFAR-10.

**Computational cost during evaluation time**  Since local Lipschitz bound needs to be evaluated for every input and global Lipschitz bound does not depend on the input, our method involves additional computation cost during certification. Let $u(t)$ be the singular vector computed by power iteration at iteration $t$, we stop power iteration when $\|u(t+1) - u(t)\| \leq 1e{-}3$. To analyze the computational cost during evaluation time, we plot the histogram of number of iterations for power method to converge for the second last layer in the 6C2F model in Figure 4. The average number of iterations for convergence is 137. The histograms for other layers are in Appendix C. To reduce the computational cost, we only need to compute local Lipschitz bound for samples that cannot be certified by global Lipschitz bound or cannot be attacked by adversaries. The proportion of those samples is $(100\% - \text{PGD Err} - \text{Global Certified Acc})$. For the 6C2F model on CIFAR-10, the proportion of samples that can be certified using global Lipschitz bound is $51.0\%$, and the error under PGD attack is $35.2\%$. Hence we only need to evaluate local Lipschitz bounds on the remaining $13.8\%$ samples, which greatly reduces the overhead of computing the local Lipschitz bounds.

| Method | Model | Clean (%) | PGD (%) | Certified(%) |
|---|---|---|---|---|
| **MNIST** ($\epsilon = 1.58$) | | | | |
| Standard | 4C3F | 99.0 | 45.4 | 0.0 |
| LMT [19] | 4C3F | 86.5 | 53.6 | 40.5 |
| CAP [21] | 4C3F | 88.1 | 67.9 | 44.5 |
| CROWN-IBP[*] [14] | 4C3F | 82.3 | 80.4 | 41.3 |
| GloRo [4] | 4C3F | 92.9 | 68.9 | 50.1 |
| Local-Lip-G (ours) | 4C3F | **96.3** | **78.2** | **55.8** |
| BCP [20] | 4C3F | 92.4 | 65.8 | 47.9 |
| Local-Lip-B (ours) | 4C3F | 93.0 | 66.7 | 48.7 |
| **CIFAR-10** ($\epsilon = {}^{36}/_{255}$) | | | | |
| Standard | 4C3F | 85.3 | 41.2 | 0.0 |
| IBP [12] | 4C3F | 34.5 | 31.8 | 24.4 |
| LMT [19] | 4C3F | 56.5 | 49.8 | 37.2 |
| CAP [21] | 4C3F | 60.1 | 55.7 | 50.3 |
| CROWN-IBP[*] [14] | 4C3F | 54.2 | 52.7 | 41.9 |
| ReLU-Stability[†] [33] | 4C3F | 57.4 | 52.4 | 51.1 |
| GloRo [4] | 4C3F | 73.2 | 66.3 | 49.0 |
| Local-Lip-G (ours) | 4C3F | **75.7** | **68.6** | 49.7 |
| BCP [20] | 4C3F | 64.4 | 59.4 | 50.0 |
| Local-Lip-B (ours) | 4C3F | 70.1 | 64.2 | **53.5** |
| Standard | 6C2F | 87.5 | 32.5 | 0.0 |
| IBP [12] | 6C2F | 33.0 | 31.1 | 23.4 |
| LMT [19] | 6C2F | 63.1 | 58.3 | 38.1 |
| CAP [21] | 6C2F | 60.1 | 56.2 | 50.9 |
| CROWN-IBP[*] [14] | 6C2F | 53.7 | 52.2 | 41.9 |
| GloRo [4] | 6C2F | 70.7 | 63.8 | 49.3 |
| Local-Lip-G (ours) | 6C2F | **76.4** | **69.2** | 51.3 |
| BCP [20] | 6C2F | 65.7 | 60.8 | 51.3 |
| Local-Lip-B (ours) | 6C2F | 70.7 | 64.8 | **54.3** |
| GloRo + MaxMin [4] | 6C2F | 77.0 | 69.2 | 58.4 |
| Caylay + MaxMin [30] | 6C2F | 75.3 | 67.7 | 59.2 |
| Local-Lip-B + MaxMin (ours) | 6C2F | **77.4** | **70.4** | **60.7** |
| **Tiny-Imagenet** ($\epsilon = {}^{36}/_{255}$) | | | | |
| Standard | 7C1F | 35.9 | 19.4 | 0.0 |
| GloRo [4] | 7C1F | 31.3 | 28.2 | 13.2 |
| Local-Lip-G (ours) | 8C2F | **37.4** | **34.2** | 13.2 |
| BCP [20] | 8C2F | 28.7 | 26.6 | 20.0 |
| Local-Lip-B (ours) | 8C2F | 30.8 | 28.4 | **20.7** |
| Gloro + MaxMin [4] | 8C2F | 35.5 | 32.3 | 22.4 |
| Local-Lip-B + MaxMin (ours) | 8C2F | **36.9** | **33.3** | **23.4** |

[*] CROWN-IBP was originally designed for $\ell_\infty$ norm certified defense but its released code also supports $\ell_2$ training. We use the same hyperparameters as $\ell_\infty$ training setting.

[†] [33] was designed for $\ell_\infty$ norm with a MIP verifier. We extend it to the $\ell_2$ norm setting and verify its robustness using the SOTA alpha-beta-CROWN verifier (see Section C.2).

Table 2: Comparison to other certified training algorithms. Best numbers are highlighted in bold.

## 5 Conclusion

In this work, we propose an efficient way to incorporate local Lipschitz bound for training certifiably robust neural networks. We classify the outputs of activation function as being constant or varying under input perturbations and consider them separately. Specifically, we remove the redundant rows and columns corresponding to the constant activation outputs in the weight matrix to get a tighter local Lipschitz bound. We propose a learnable bounded activation and the use of a sparsity encouraging loss during training to assist the neural network to learn to tighten our local Lipschitz bound. Our method consistently outperforms state-of-the-art Lipschitz based certified defend methods for $\ell_2$ norm robustness. We see no immediate negative societal impact in the proposed approach.

**Acknowledgements**

Y. Huang is supported by DARPA LwLL grants. A. Anandkumar is supported in part by Bren endowed chair, DARPA LwLL grants, Microsoft, Google, Adobe faculty fellowships, and DE Logi grant. Huan Zhang is supported by funding from the Bosch Center for Artificial Intelligence.

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
