# Appendix

## A   Method Details

### A.1   A Toy example

In this section, we provide a toy example to walk through the tighter local Lipschitz bound calculation method in a three-layer neural network step by step. Consider a 3-layer neural network with ReLU$\theta$ activation:

$$x \to W^1 \to \text{ReLU}\theta \to W^2 \to \text{ReLU}\theta \to W^3 \to y\,,$$

where input $x \in \mathbb{R}^3$ and output $y \in \mathbb{R}$.

Suppose at a certain training epoch $t$, weight matrices have the following values,

$$W^1 = W^2 = \begin{bmatrix} 3 & 0 & 0 \\ 0 & 2 & 0 \\ 0 & 0 & 1 \end{bmatrix}, W^3 = \begin{bmatrix} 1 & 1 & 1 \end{bmatrix}, \theta = 1.$$

Consider an input $x = [1, -1, 0]^\top$ and input perturbation $||x' - x|| \le 0.1$. Directly using the global Lipschitz bound, we have the following result,

$$L_{glob} \le ||W^3||\,||W^2||\,||W^1|| = 9\sqrt{3}. \tag{13}$$

Consider our approach for local Lipschitz bound computation. Given input $x = [1, -1, 0]^\top$ and bounded perturbed input $||x' - x|| \le 0.1$, the feature map after $W_1$ is $[[2.7, 3.3], [-2.2, -1.8], [-0.1, 0.1]]^\intercal$, where we use $[\text{LB}, \text{UB}]$ to denote the interval bound for every entry. After ReLU$\theta$, the feature map turns into $[1, 0, [0, 0.1]]^\intercal$, where the first entry is a constant because it is clipped by the upper threshold $\theta$, and the second entry is a constant because it is always less than zero. The third entry varies under perturbation. Using these interval bounds, we can compute the indicator matrices $I_C^1$, $I_\theta^1$ and $I_V^1$. After the second weight matrix $W_2$, the feature map is $[3, 0, [0, 0.1]]^\intercal$. We can compute the local indicator matrices at this layer accordingly. Specifically, the local indicator matrices are as follows,

$$I_C^1 = \begin{bmatrix} 0 & 0 & 0 \\ 0 & 1 & 0 \\ 0 & 0 & 0 \end{bmatrix}, I_\theta^1 = \begin{bmatrix} 1 & 0 & 0 \\ 0 & 0 & 0 \\ 0 & 0 & 0 \end{bmatrix}, I_V^1 = \begin{bmatrix} 0 & 0 & 0 \\ 0 & 0 & 0 \\ 0 & 0 & 1 \end{bmatrix},$$

$$I_C^2 = \begin{bmatrix} 0 & 0 & 0 \\ 0 & 1 & 0 \\ 0 & 0 & 0 \end{bmatrix}, I_\theta^2 = \begin{bmatrix} 1 & 0 & 0 \\ 0 & 0 & 0 \\ 0 & 0 & 0 \end{bmatrix}, I_V^2 = \begin{bmatrix} 0 & 0 & 0 \\ 0 & 0 & 0 \\ 0 & 0 & 1 \end{bmatrix},$$

And the local Lipschitz bound around input $x = [1, -1, 0]^\top$ follows,

$$\begin{aligned} L_{\text{local}}(x) &\le ||W^3 I_V^2||\,||I_V^2 W^2 I_V^1||\,||I_V^1 W^1|| \\ &= ||\begin{bmatrix} 0 & 0 & 1 \end{bmatrix}||\,||\begin{bmatrix} 0 & 0 & 0 \\ 0 & 0 & 0 \\ 0 & 0 & 1 \end{bmatrix}||\,||\begin{bmatrix} 0 & 0 & 0 \\ 0 & 0 & 0 \\ 0 & 0 & 1 \end{bmatrix}|| = 1 \end{aligned} \tag{14}$$

which is significantly tighter than the global Lipschitz bound obtained in Eq (13).

### A.2   Why not consider linear ReLU outputs?

Our approach for tighter local Lipschitz bound separate ReLU outputs to two classes: constant ReLU outputs and varying ReLU outputs under perturbation. In fact, there is another case of ReLU outputs, where the outputs of ReLU equal to the inputs. We refer to these outputs as linear ReLU outputs. In this section, we derive the local Lipschitz bound when considering linear ReLU outputs, and we will find this bound is not necessarily smaller than the global Lipshitz bound.

For simplicity, let us consider the standard ReLU activation. There are three different types of ReLU outputs (linear, fixed, varying). We inherit the notations from the main text, and use $I_L$ to denote

the indicator matrix for linear ReLU outputs. The indicator matrices for the three different types of ReLU outputs are:

$$I_{\mathrm{L}}^l(i,i) = \begin{cases} 1 & \text{if } \mathrm{LB}_i^l > 0 \\ 0 & \text{otherwise} \end{cases}, \quad I_{\mathrm{C}}^l(i,i) = \begin{cases} 1 & \text{if } \mathrm{UB}_i^l \leq 0 \\ 0 & \text{otherwise} \end{cases}, \quad I_{\mathrm{V}}^l = I - I_{\mathrm{L}} - I_{\mathrm{C}}, \quad (15)$$

where $I$ is the identity matrix.

In addition, we use $D_{\mathrm{L}}$ and $D_{\mathrm{V}}$ to denote the slope of ReLU outputs for the linear and varying neurons. The elements in $D_{\mathrm{L}}$ and $D_{\mathrm{V}}$ can either be 0 or 1 due to the piece-wise linear property of ReLU. We omit the constant ReLU outputs here because they are all zeros.

$$D_{\mathrm{L}}^l(i,i) = \begin{cases} 1 & \text{if } I_{\mathrm{L}}^l(i,i) = 1 \\ 0 & \text{if } I_{\mathrm{L}}^l(i,i) = 0 \end{cases}, \quad D_{\mathrm{V}}^l(i,i) = \begin{cases} \mathbb{1}(\mathrm{ReLU}(z_i^l) > 0) & \text{if } I_{\mathrm{V}}^l(i,i) = 1 \\ 0 & \text{if } I_{\mathrm{V}}^l(i,i) = 0 \end{cases}, \quad (16)$$

where $\mathbb{1}$ denotes an indicator function.

Let us consider a 3-layer neural network with ReLU activation:

$$x \rightarrow W^1 \rightarrow \mathrm{ReLU} \rightarrow W^2 \rightarrow \mathrm{ReLU} \rightarrow W^3 \rightarrow y.$$

Then, the neural network function (denoted as $F(x; W)$) can be written as:

$$\begin{aligned} F(x; W) &= (W^3(D_{\mathrm{L}}^2 + D_{\mathrm{V}}^2)W^2(D_{\mathrm{L}}^1 + D_{\mathrm{V}}^1)W^1)x \\ &= (W^3 D_{\mathrm{L}}^2 W^2 D_{\mathrm{L}}^1 W^1 + W^3 D_{\mathrm{L}}^2 W^2 D_{\mathrm{V}}^1 W^1 + W^3 D_{\mathrm{V}}^2 W^2 D_{\mathrm{L}}^1 W^1 + W^3 D_{\mathrm{V}}^2 W^2 D_{\mathrm{V}} W^1)x \end{aligned} \quad (17)$$

Bounding the local Lipschitz constant from Eq (17), we get

$$\begin{aligned} L_{\mathrm{local}}' &= \|W^3 D_{\mathrm{L}}^2 W^2 D_{\mathrm{L}}^1 W^1 + W^3 D_{\mathrm{L}}^2 W^2 D_{\mathrm{V}}^1 W^1 + W^3 D_{\mathrm{V}}^2 W^2 D_{\mathrm{L}}^1 W^1 + W^3 D_{\mathrm{V}}^2 W^2 D_{\mathrm{V}}^1 W^1\| \\ &\leq \|W^3 D_{\mathrm{L}}^2 W^2 D_{\mathrm{L}}^1 W^1\| + \|W^3 D_{\mathrm{L}}^2 W^2 D_{\mathrm{V}}^1 W^1\| + \|W^3 D_{\mathrm{V}}^2 W^2 D_{\mathrm{L}}^1 W^1\| + \|W^3 D_{\mathrm{V}}^2 W^2 D_{\mathrm{V}}^1 W^1\| \end{aligned} \quad (18)$$

$$\begin{aligned} &\leq \|W^3 D_{\mathrm{L}}^2 W^2 D_{\mathrm{L}}^1 W^1\| + \|W^3 D_{\mathrm{L}}^2 W^2 I_{\mathrm{V}}^1\|\|I_{\mathrm{V}}^1 W^1\| + \|W^3 I_{\mathrm{V}}^2\|\|I_{\mathrm{V}}^2 W^2 D_{\mathrm{L}}^1 W^1\| \\ &\quad + \|W^3 I_{\mathrm{V}}^2\|\|I_{\mathrm{V}}^2 W^2 I_{\mathrm{V}}^1\|\|I_{\mathrm{V}}^1 W^1\|, \end{aligned} \quad (19)$$

where Eq (18) uses triangular inequality, and Eq (19) uses Cauchy–Schwarz inequality and the fact that the Lipschitz constant of ReLU is smaller than 1.

The key observation from this approach is that we can "merge" the weight matrices together for linear neurons (the first term in Eq (19)). Then we have $\|W^3 D_{\mathrm{L}}^2 W^2 D_{\mathrm{L}}^1 W^1\| \leq \|W^3\|\|W^2\|\|W^1\|$. Furthermore, if the singular vectors for the weight matrices are not aligned, $\|W^3 D_{\mathrm{L}}^2 W^2 D_{\mathrm{L}}^1 W^1\|$ will be much tighter than $\|W^3\|\|W^2\|\|W^1\|$.

However, there are three other non-negative terms in Eq (19) from the triangular inequality. Even though the first term could be smaller than the global Lipschitz bound, the summation can be larger than the global Lipschitz bound. In comparison, our approach always guarantees tighter Lipschitz bound as proved in Theorem 1.

## B  Review of other robust training methods

During training, we combine our method with state-of-the-art certifiable training algorithms that involves using L2 Lipschitz bound such as BCP [20] and Gloro [4]. In this section, we first introduce the goal of certifiable robust training and then describe BCP and Gloro in more detail.

Consider a neural network that maps input $x$ to output $z = F(x)$, where $z \in \mathbb{R}^N$. The ground truth label is $y$. The goal of certifiable training is to minimize the the certified error $R$. However, computing the exact solution of $R$ is NP-complete [9]. So in practice, many certifiable training algorithms compute an outer bound of the perturbation sets in logit space $\hat{z}(\mathbb{B}(x))$, and find the worst logit $z^* \in \hat{z}(\mathbb{B}(x))$ to obtain an upper bound of $R$,

$$\begin{aligned} R &= \mathbb{E}_{(x,y)\sim\mathcal{D}}[\max_{z \in F(\mathbb{B}_2(x,\epsilon))} \mathbb{1}(\arg\max z \neq y)] \\ &\leq \mathbb{E}_{(x,y)\sim\mathcal{D}}[\max_{z \in \hat{z}(\mathbb{B}(x))} \mathbb{1}(\arg\max z \neq y)] \\ &= \mathbb{E}_{(x,y)\sim\mathcal{D}}[\mathbb{1}(\arg\max z^* \neq y)] \end{aligned} \quad (20)$$

where $\mathbb{B}_2(x, \epsilon)$ denotes the $\ell_2$ perturbation set in the input space, $\mathbb{B}_2(x, \epsilon) = \{x' : \|x' - x\|_2 \le \epsilon\}$, and $F(\mathbb{B}_2(x, \epsilon)) \subset \hat{z}(\mathbb{B}(x))$. In the subsequent text, we use $\mathbb{B}(x)$ in place of $\mathbb{B}_2(x, \epsilon)$ for short. The main difference between BCP and Gloro is how they compute $\hat{z}(\mathbb{B}(x))$ and find the worst logits.

**BCP**    In BCP, the perturbation set in the input space is propagated through all layers except the last one to get the ball outer bound $\mathbb{B}_2^l$ and box outer bound $\mathbb{B}_\infty^l$ at layer $l$. Specifically, the $l$-th layer box outer bound with midpoint $m^l$ and radius $r^l$ is $\mathbb{B}_\infty^l(m^l, r^l) = \{z^l : |z_i^l - m_i^l| \le r_i^l, \forall i\}$. To compute the indicator matrix in Equation (3) and (9) for our local Lipschitz bound, we need to bound each neuron by $lb_i^l \le z_i^l \le ub_i^l$, where

$$ub^l = \min(m_\infty^l + r_\infty^l, m_{ball}^l + r_{ball}^l) \tag{21}$$

$$lb^l = \max(m_\infty^l - r_\infty^l, m_{ball}^l - r_{ball}^l) \tag{22}$$

In the case of linear layers, it follows

$$m_\infty^l = W^l m^{l-1} + b^l, r_\infty^l = |W^l| r^{l-1}; m_{ball}^l = W^l z^{l-1} + b^l, r_{ball}{}_i^l = \|W_{i,:}^l\| \rho^{l-1} \tag{23}$$

where $\rho^{l-1} = \epsilon \prod_{k=1}^{l-1} \|W^k\|, m^{l-1} = (ub^{l-1} + lb^{l-1})/2$, and $r^{l-1} = (ub^{l-1} - lb^{l-1})/2$.

Then an outer bound of the perturbation sets in logit space $\hat{z}(\mathbb{B}(x))$ is computed via the ball and box constraints on the second last layer: $\hat{z}(\mathbb{B}(x)) = W^L(\mathbb{B}_\infty^{L-1} \cap \mathbb{B}_2^{L-1})$. Finally, the worst logit is computed as

$$z_i^* = F_y(x) - \min_{z \in \hat{z}(\mathbb{B}(x))} (z_y - z_i) \tag{24}$$

where $F_y(x)$ denotes the $y$ th entry of $F(x)$ that corresponds to the ground truth label.

**Gloro**    Unlike BCP, Gloro does not use the local information from box propagation to compute $\hat{z}(\mathbb{B}(x))$ for computation efficiency. In addition, Gloro creates a new class in the logits indicating non-certifiable prediction. The worst logit is computed as by appending the new entry after original logits output $\tilde{z}(x) = [F(x)| \max_{m \ne y} z_m^*]$, where $z_m^*$ is computed by (24) with only ball constraints $\hat{z}(\mathbb{B}(x)) = W^L(\mathbb{B}_2^{L-1})$. However, when we combine our method with Gloro, we use their way to construct the new class in the worst logit, but keeps the box constraint when computing $\hat{z}(\mathbb{B}(x))$.

The loss in certifiable training algorithms is a mixed loss function on a normal logit $z = F(x)$ and the worst logit $z^*(x)$:

$$\mathcal{L} = \mathbb{E}_{(x,y) \sim \mathcal{D}}[(1 - \lambda)\mathcal{L}(z(x), y) + \lambda \mathcal{L}(z^*(x), y)] \tag{25}$$

where $\mathcal{L}$ denotes cross entropy loss, and $\lambda$ is a hyper-parameter.

## C    Experimental Details

### C.1    Training details

**Computing Resources**    We train our MNIST and CIFAR models on 1 NVIDIA V100 GPU with 32 GB memory. We train our Tiny-Imagenet model on 4 NVIDIA V100 GPUs.

**Architecture**    We denote a convolutional layer with output channel $c$, kernel size $k$, stride $s$ and output padding $p$ as $C(c, k, s, p)$ and the fully-connected layer with output channel $c$ as $F(c)$. We apply ReLU$\theta$ activation after every convolutional layer and fully-connected layer except the last fully-connected layer.

- 4C3F: C(32,3,1,1)-C(32,4,2,1)-C(64,3,1,1)-C(64,4,2,1)-F(512)-F(512)-F(10)

- 6C2F: C(32,3,1,1)-C(32,3,1,1)-C(32,4,2,1)-C(64,3,1,1)-C(64,3,1,1)-C(64,4,2,1)-F(512)-F(10)

- 8C2F: C(64,3,1,1)-C(64,3,1,1)-C(64,4,2,0)-C(128,3,1,1)-C(128,3,1,1)-C(128,4,2,0)-C(256,3,1,1)-C(256,4,2,0)-F(256)-F(200)

| Dataset | MNIST | CIFAR | Tiny-Imagenet (ReLU) | Tiny-Imagenet (MaxMin) |
|---|---|---|---|---|
| Initial LR | 0.001 | 0.001 | 2.5e−4 | 1e−4 |
| End LR | 5e−6 | 1e−6 | 5e−7 | 5e−7 |
| Batch Size | 256 | 256 | 128 | 128 |
| $\epsilon_{\text{train}}$ | 1.58 | 0.1551 | 0.16 | 0.16 |
| $\lambda_{\text{sparse}}$ | 0.0 | 0.0 | 0.01 | 0.01 |
| $\lambda_\theta$ | 0.0 | 0.0 | 0.1 | 0.1 |
| Warm-up Epochs | 0 | 20 | 0 | 0 |
| Total Epochs | 300 | 800 | 250 | 250 |
| LR Decay Epoch ($m$) | 150 | 400 | 150 | 150 |
| $\epsilon$ Sched. Epochs ($n$) | 150 | 400 | 125 | 125 |
| Power Iters. | 5 | 2 | 1 | 3 |

Table 3: Hyper-parameters used in certifiable training.

**Loss** We train with the standard certifiable training loss from Eq (25) and the sparsity loss from Eq (12) used to encourage constant neurons. The total loss for ReLU networks is:

$$\mathcal{L} = \mathbb{E}_{(x,y)\sim\mathcal{D}}[(1-\lambda)\mathcal{L}(z(x),y) + \lambda\mathcal{L}(z^*(x),y)] + \lambda_{\text{sparsity}}(\max(0, \text{UB}_i^l) + \lambda_\theta \max(0, \theta_i - \text{LB}_i^l)) \tag{26}$$

where $\lambda$, $\lambda_{\text{sparsity}}$ and $\lambda_\theta$ are hyper-parameters. The total loss for MaxMin networks is defined similarly with $\mathcal{L}_{\text{sparsity}}^{\text{MaxMin}}$ from Eq (12).

**Hyper-parameters** The hyper-parameters that we use during training is listed in Table 3. Power iters. stands for the number of power iterations that we use during training. Intial threshold for ReLU$\theta$ is the initialization value of the upper threshold in ReLU$\theta$ . For all our experiments, we use the Adam optimizer [34]. For CIFAR experiments, we use the same hyper-parameters for both ReLU and MaxMin activations.

*Learning rate scheduling* We train with the initial learning rate for $m$ epochs and then start exponential learning rate decay. Let $T$ be the total number of epochs. Learning rate (LR) for epoch $t$ is:

$$\text{LR}(t) = \begin{cases} \text{Initial\_LR} & \text{if } t \le m \\ \text{Initial\_LR}\left(\frac{\text{End\_LR}}{\text{Initial\_LR}}\right)^{\frac{t-m}{T-m}} & \text{if } t > m \end{cases} \tag{27}$$

$\epsilon$ *scheduling* We gradually increase $\epsilon$ during training to a target value $\epsilon_{target}$ over $n$ epochs. The target value is set to be slightly larger than the $\epsilon$ that we aim to certify during evaluation time to give better performance [4]. $\epsilon$ for epoch $t$ is:

$$\epsilon(t) = \begin{cases} \frac{t}{n}\epsilon_{target} & \text{if } t \le n \\ \epsilon_{target} & \text{if } t > n \end{cases} \tag{28}$$

*mixture loss scheduling* When combined with BCP, we train with the mixed loss of clean cross entropy loss and robust cross entropy loss. We increase $\lambda$ in Equation 25 from 0 to 1 linearly over $n$ epochs. $\lambda$ for epoch $t$ is:

$$\lambda(t) = \begin{cases} \frac{t}{n} & \text{if } t \le n \\ 1 & \text{if } t > n \end{cases} \tag{29}$$

## C.2 Ablation Studies

**Reproducibility** We train each model with 3 random seeds and report the average accuracy and the standard deviation in Table 4. For MNIST, we train with our local Lipscthiz bound and the Gloro loss. CIFAR and Tiny-Imagenet, we train with our local Lipscthiz bound and the BCP loss. In the main text (Table 2), we report the best accuracy out of 3 runs.

**Comparison to techniques designed to certify a fixed, post-training network** We used the state-of-the-art NN verifier (alpha-beta-CROWN [28, 35, 36]) from the VNN challenge [37] to certify a fixed, post-training network trained. The trained network is trained by techniques proposed

| Data | $\epsilon$ | Model | Clean (%) | PGD (%) | Certified (%) |
|---|---|---|---|---|---|
| MNIST | 1.58 | 4C3F | $96.29 \pm 0.07$ | $78.31 \pm 0.08$ | $55.79 \pm 0.23$ |
| CIFAR-10 | 36/255 | 4C3F | $69.78 \pm 0.30$ | $63.95 \pm 0.26$ | $53.40 \pm 0.08$ |
| CIFAR-10 | 36/255 | 6C2F | $70.64 \pm 0.05$ | $64.81 \pm 0.05$ | $53.96 \pm 0.60$ |
| Tiny-Imagenet | 36/255 | 8C2F | $29.78 \pm 0.08$ | $27.7 \pm 0.13$ | $20.5 \pm 0.13$ |

Table 4: Average accuracy and standard deviation over 3 runs. The performance of our method is consistent across different runs.

by Xiao et al. [33]. Xiao et al. [33] proposed a $\ell_\infty$ norm certified defense by imposing ReLU stability regularizer with adversarial training, and verifying the network using a mixed integer linear programming (MILP) verifier. The original approach in Xiao et al. [33] is not directly applicable to our setting, as they relied on the MILP verifier which cannot scale to the large models evaluated in our paper, and they focused on $\ell_\infty$ norm robustness. To make a fair comparison to their approach, we made a few extensions to their paper:

- We use $\ell_2$ norm adversarial training to replace their $\ell_\infty$ norm adversarial training.
- We use the same large CIFAR network (4C3F with 62464 neurons) as in our other experiments.
- We use the best NN verifier in the very recent VNN COMP 2021 [37], alpha-beta-CROWN [28, 36], to replace their MILP based verifier.

Additionally, we tried different regularization parameters and reported the best results here. The clean accuray is $57.39\%$, PGD accuracy is $52.41\%$ and the verified Accuracy is $51.09\%$. This approach produces a reasonably robust model, thanks to the very recent strong NN verifier. However, its clean, verified and PGD accuracy are worse than ours. Additionally this approach is much less scalable than ours - the verification takes about 150 GPU hours to finish, while our approach takes only 7 minutes to verify the entire dataset.

**Comparison to adversarial training** We compared our method with adversarial training (AT) [6] and TRADES [38] methods as these are known to regularize the Lipschitz constant of neural networks and provide robustness. Although AT and TRADES regularize the neural network Lipschitz compared to a naturally trained neural network, it is still not enough to provide certified robustness. We train with AT and TRADES on CIFAR-10 with the 6C2F architecture and report their accuracies, global and local Lipschitz bounds in Table 5. The certified accuracy is calculated using local Lipschitz bound. As we can see from the table, although the models trained via AT and TRADES have much smaller Lipschitz bound than naturally trained models (we only use cross entropy loss on clean images in natural training), the Lipschitz bound is still too large to give certified robustness. We also used alpha-beta-CROWN to certify the adversarially trained CIFAR-10 4C3F network. The verified accuracy is still $0\%$.

| Methods | Global Lipschitz bound | Local Lipschitz bound | Clean (%) | PGD (%) | Certified (%) |
|---|---|---|---|---|---|
| Standard | $1.52 \times 10^9$ | $4.81 \times 10^8$ | 87.7 | 35.9 | 0.0 |
| AT | $2.27 \times 10^4$ | $1.85 \times 10^4$ | 80.7 | 70.76 | 0.0 |
| TRADES | $1.82 \times 10^4$ | $1.32 \times 10^4$ | 80.0 | 72.3 | 0.0 |
| BCP | 11.35 | 11.08 | 65.7 | 60.8 | 51.3 |
| Ours | 7.89 | 6.68 | 70.7 | 64.8 | 54.3 |

Table 5: Comparison to adversarial training methods on CIFAR-10 with the 6C2F architecture.

**Influence of sparsity loss on certified robustness** To encourage the sparsity of varying ReLU neurons, we design a hinge loss to regularize the neural network (Eq 26). To study the effectiveness of this sparsity loss, we vary the coefficient of this loss when training a 8C2F model on Tiny-Imagenet. In addition, we find that down-weighting the hinge loss on the upper threshold improves performance. Hence we keep $\lambda_\theta$ as 0.1 for all the models. We report the clean, PGD and certified accuracy in Table 6. As we can see, too large coefficients on the sparsity loss tends to over-regularize the neural network. When $\lambda_{\text{sparse}} = 0.01$, we obtain the best performance.

| $\lambda_{\text{sparse}}$ | Clean (%) | PGD (%) | Certified (%) |
|---|---|---|---|
| 0.0 | 30.6 | 28.3 | 19.9 |
| 0.01 | 30.8 | 28.4 | 20.7 |
| 0.03 | 29.7 | 27.3 | 20.2 |
| 0.1 | 28.9 | 26.7 | 19.8 |

Table 6: Influence of sparsity loss on certified robustness on the TinyImageNet dataset with the 8C2F model.

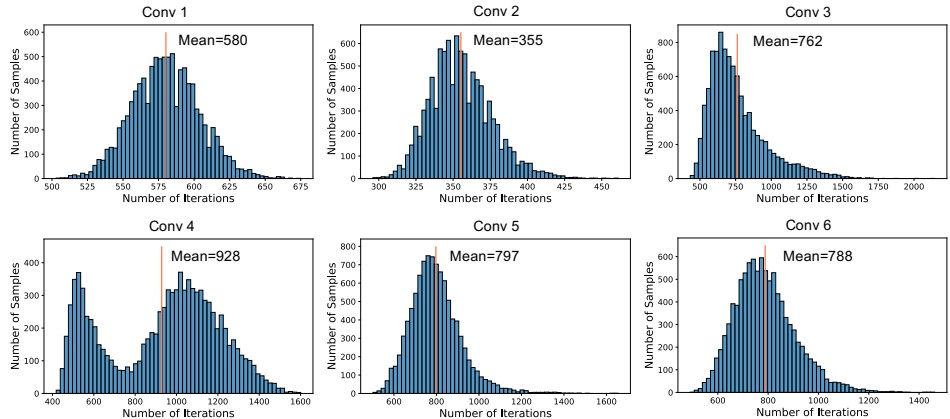

Figure 5: Number for power method to converge at each convolutional layer in the 6C2F model.

**Time cost in training**  Since our method needs to evaluate local Lipschitz bound on every data point during training, we pay additional computational cost than Gloro [4] and BCP [20]. However, our local Lipschitz bound is still much more computational efficient than convex relaxation methods such as CAP [22]. We report the computation time (s/epoch) in Table 7.

| | | Computation time (s/epoch) | | | |
|---|---|---|---|---|---|
| Data | Model | CAP | Gloro | BCP | Ours |
| MNIST | 4C3F | 689 | 9.0 | 17.3 | 45.5 |
| CIFAR-10 | 4C3F | 645 | - | 23.5 | 38.2 |
| CIFAR-10 | 6C2F | 1369 | 6.5 | 26.0 | 69.8 |
| Tiny-Imagenet | 8C2F | - | - | 343.5 | 398.8 |

Table 7: Comparison of training time per epoch.

**Number of power iterations for convergence**  To analyze the computational cost, we plot the histogram of number for power method to converge at each convolutional layer in Figure 5. We used the 6C2F model on CIFAR-10. Let $u(t)$ be the singular vector computed by power iteration at iteration $t$, we stop power iteration when $\|u(t+1) - u(t)\| \leq 1\mathrm{e}-3$.

**Lipschitz bounds and Sparsity of varying ReLU outputs during training**  We track the global and Local Lipschitz bound during training for a standard trained CNN, a CNN trained with BCP and global Lipschitz bound, and a CNN trained with BCP and our local Lipschitz bound. All the models are trained with the 6C2F architecture on CIFAR-10. For BCP and our method, we train with $\epsilon_{\text{train}} = 36/255$. We used the hyper-parameters found by BCP [20] to train. The total epochs is 200, and first 10 epochs are used for warm-up. We decay learning rate by half every 10 epochs starting from the 121-th epoch. The Lipschitz bound change during training is shown in Figure 7. Meanwhile, we track the proportion of varying ReLU outputs for all the layers during training in Figure 6. We can see that the models trained for certified robustness have much fewer varying ReLU outputs than the standard trained model. The sparsity of varying ReLU outputs is desired to tighten our local Lipschitz bound since we can remove more redundant rows and columns in weight matrices.

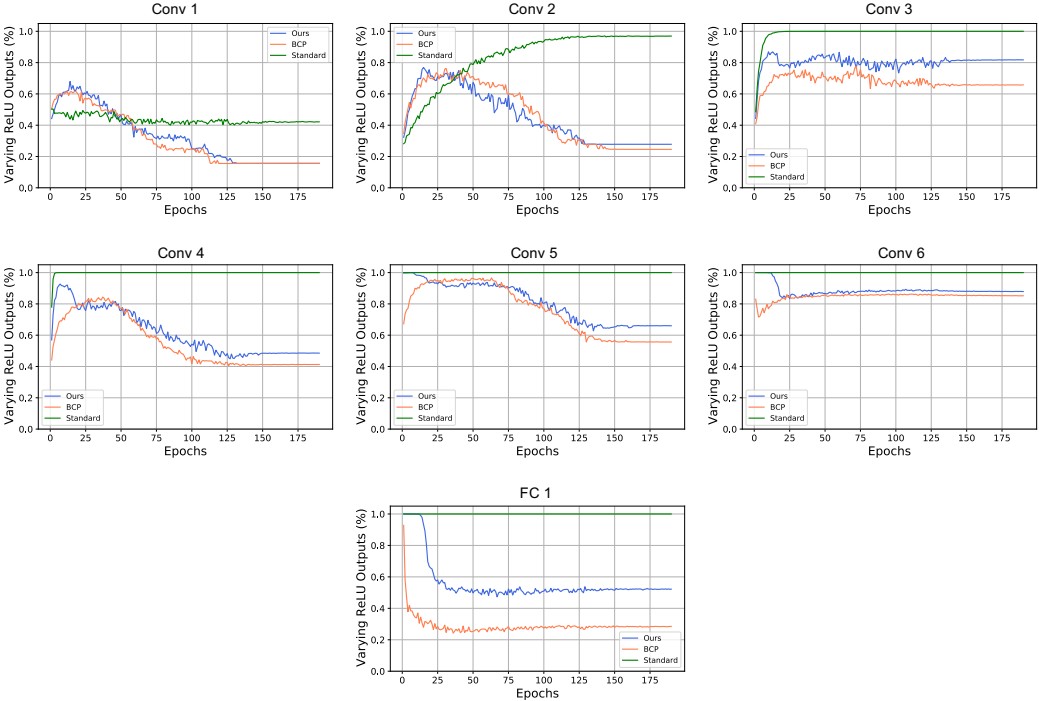

Figure 6: Proportion of varying ReLU outputs for all the layers in 6C2F model during training.

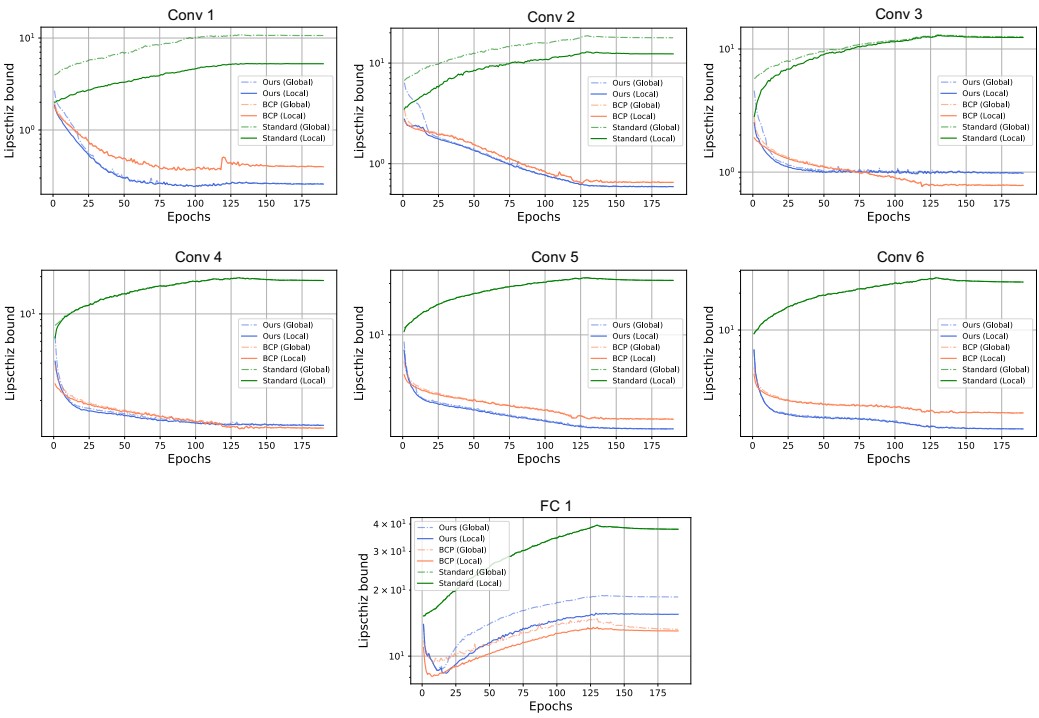

Figure 7: Lipschitz bound for all the layers in 6C2F model during training.