# OpenReview forum: "Training Certifiably Robust Neural Networks with Efficient Local Lipschitz Bounds"
_NeurIPS.cc/2021/Conference — NeurIPS 2021 Poster_

### Official Review · Reviewer_UGUy · 2021-07-08

**Rating:** 6
**Confidence:** 3

**Summary:**

This paper studies certified robustness from the perspective of computing Lipschitzness bounds. Prior work compute global bounds on the Lipschitz constant of the network; this paper takes an alternative approach which computes local Lipschitz bounds around a specific input. These local Lipschitz bounds are based on interval bounds for the hidden layers on each data point, and computing a tighter Lipschitz constant based on eliminating rows/columns of the weight matrices when the interval bounds indicate that a neuron is inactive.

To train certifiably robust models, the worst-case loss is optimized based on these Lipschitz bounds. There are several other ingredients for the proposed method: a modified ReLU activation which truncates when the output is a large positive value, and a sparsity-inducing regularization meant to limit the number of neurons where the upper and lower activation bounds differ. The authors compare certified robustness w.r.t. $\ell_2$ perturbations against other Lipschitzness-based baselines and demonstrate improvements.

**Limitations And Societal Impact:**

Yes.

**Main Review:**

The paper proposes a certified robustness method based on nice and novel insights about training the local Lipschitz constants rather than the global constants. The method produces empirical improvements over existing methods; however, there is still some uncertainty about the effects of different components of the proposed methods, which aren't adequately addressed in the ablations.  (More details below.) For this reason, I'm recommending weak accept, but am willing to change my score based on the authors' response.

Originality: the idea of regularizing local Lipschitzness is novel and a nice way to improve over prior work.

Quality: the paper appears to be technically sound, but it would be helpful to have more experiments which isolate the effects of each component of the proposed method (see below).

Clarity: the paper is generally well-written, though there are some portions where clarification would help (see questions below.)

Significance: setting aside the above concerns, this paper makes nice contributions to certified robustness.

More details about questions/ concerns:
- First, regarding the different components of the objective -- it would be nice to see how much of the improvement is due to the sparsity-inducing loss and how much is due to regularizing the local Lipschitz constant. Table 5 in the supplementary material does perform an ablation on this, but when $\lambda = 0$ (no sparsity loss), the reported numbers are actually worse than the reported numbers for BCP in Table 1. This makes me concerned that a lot of the improvement might actually come from including the sparsity loss, not regularizing the local Lipschitz constant.

Another concern is how much the change in architecture accounts for the differences in performance -- it would be nice to see an ablation study on the impact of the modified ReLU activation.

-- It's also unclear how exactly the local lipschitz constant is optimized. E.g. is the power iteration unrolled, and backprop is performed through the power iterations?

Some other questions/clarifications:
-- The definition of D_V in (5) doesn't appear completely right -- shouldn't it be a binary diagonal matrix?
-- For equation (6), would it also be possible to bound the Lipschitz constant by $\|\prod_l I^l W^l I^{l - 1} \|$ instead of the product of the norms? This seems like it could lead to a tighter bound.

=======================================================================
Update after reading the author responses and other reviews. While I appreciate and thank the authors for their thorough response to my questions, I think this paper remains borderline, especially given some of the concerns raised by other reviewers. Thus, I'm keeping my score the same.

**Time Spent Reviewing:**

3

---

> ### Author Response · Authors · 2021-08-10
> **Response to reviewer UGUy**
>
> We thank the reviewer for carefully reviewing our manuscript and providing constructive comments. Studying the impact of each component will indeed help us better understand the effectiveness of the proposed method.
>
> **Q: Impact of the sparsity loss.**
>
> **A: Sparsity loss will only improve performance when local information is exploited.**
> Sparsity loss encourages the network to have more constant output neurons. However, if we only compute the global Lipschitz bound, the constant neurons will not benefit us because it does not influence the global Lipschitz bound of weight matrices.
> For the Tiny-imagenet results, although the certified accuracy is slightly lower than that of BCP (by ~0.1%) without sparsity loss, the clean and PGD accuracy is still much higher (by ~2%). So this still supports our claim that local Lipschitz bound adds less constraints to the neural network, so the network has larger capacity to reach higher clean accuracy.
> We also tried training with BCP and sparsity loss without local Lipschitz bound, and interestingly, sparsity loss also improves the results for BCP a bit (mainly for clean and PGD accuracy). We think this is because the sparsity loss helps make the box propagation to be tighter in BCP.
> If we further apply our local Lipschitz bound, we can achieve improvements in all clean, PGD and certified accuracy.
>
> |  Method                                                            | Clean (%) | PGD (%) | Certified (%) |
> |--------------------------------------------------------------------|-----------|---------|---------------|
> | BCP                                                                | 28.7      | 26.6    | 20.0          |
> | Local-ReLU (ours) with BCP loss                                    | 30.6      | 28.3    | 19.9          |
> | BCP + sparsity loss (coefficient=0.01)                             | 29.5      | 27.0    | 20.1          |
> | Local-ReLU (ours) with BCP loss + sparsity loss (coefficient=0.01) | **30.8**      | **28.4**    | **20.7**          |
>
>
> **Q: Impact of the modified ReLU activation.**
>
> A: We thank the reviewer for the great suggestion. **We train Gloro and BCP with ReLU_theta (the modified ReLU activation) with global Lipschitz bound, and the global Lipschitz bound gives worse certified accuracy compared with our local bound.** The accuracy is as follows.It is interesting to note that ReLU_theta does not help improving Gloro’s certified accuracy, but it does help improve BCP’s certified accuracy. We think this is because although BCP does not use global Lipschitz bound, it does use local information in box propagation, and ReLU_theta makes the local box bound to be tighter in BCP. However, our approach outperforms all baselines.
>
> |                                 | Clean (%) | PGD (%) | Certified (%) |
> |---------------------------------|-----------|---------|---------------|
> | Gloro                           | 70.7      | 63.8    | 49.3          |
> | Gloro + ReLU_theta              | 72.7      | 66.1    | 48.1          |
> | Local-ReLU_theta with Gloro loss (ours) | **76.4**      | **69.2**    | 51.3          |
> | BCP                             | 65.7      | 60.8    | 51.3          |
> | BCP + ReLU_theta                | 67.7      | 61.9    | 53.3          |
> | Local-ReLU_theta with BCP loss (ours)   | 70.7     | 64.8    | **54.3**         |
>
> **Q: How exactly is the local Lipschitz constant optimized?**
>
> A: Thanks for asking the question. Yes, we unrolled the power iteration and backprop through unrolled iterations. We will add this clarification in the revised paper.
>
> **Q: The definition of D_V in (5) doesn't appear completely right -- shouldn't it be a binary diagonal matrix?**
>
> A: We thank the reviewer for pointing out the typo. We corrected this typo in the general response.
>
> **Q: For equation (6), would it also be possible to bound the Lipschitz constant by $\prod_l I^l W^l I^{l-1}$ instead of the product of the norms? This seems like it could lead to a tighter bound.**
>
> A: We appreciate the reviewer’s insightful suggestion on directly bound the Lipschitz constant. Unfortunately, as mentioned in the general response, **computing this bound has exponentially complexity and is infeasible in practice.**  The major challenge for directly computing the spectral norm lies in the $D_V$ terms (i.e., $D_V^{L-1}, D_V^{L-2}, …, D_V^{1}$). Importantly, since $D_V$ denotes the slope of ReLU for the varying neurons, the value of D_V is not determined until the realization of local perturbations. In order to obtain a uniform upper bound, one way is to brute force compute $||(W^L I_V^{L-1}) D_V^{L-1} (I_V^{L-1} W^{L-1} I_V^{L-2} .... D_V^{1} (I_V^{1} W^{1}) x||$ for all possible realizations of $D_V^{L}$, which has $2^n$ complexity and $n$ is the feature map dimension at layer L.
>
> **Overall**, we have added more thorough ablation studies to understand the impact of each design component (sparsity loss and ReLU_theta). We also discussed the difficulties in bounding the Lipschitz of the product of matrices in deep neural nets. We thank the reviewer for helping improve the paper. If you like our paper, please champion it.

---

### Official Review · Reviewer_uE4r · 2021-07-09

**Rating:** 6
**Confidence:** 3

**Summary:**

This paper proposes using local Lipschitz bound to obtain certified robust neural networks. The use of local Lipschitz constant strictly results in a provable tighter bound than the commonly used global Lipschitz bound. To incorporate the proposed certification method into training, the authors further proposed a modified ReLU function with a learnable upper threshold, and hinge loss to encourage the pre-activation states to be constant past ReLU function.

The proposed method is sound, numerical experiments show promising improvement over the existing certified robust training methods.

**Limitations And Societal Impact:**

The authors have discussed the computational cost of the proposed training method. I think the proposed method is practical since the extra computational overhead is not very heavy.

**Main Review:**

Originality:

The use of local Lipschitz constant as certified upper bound for training robust neural networks is a novel contribution.

Quality:

The submission is technically sound. The main claim is supported by the numerical experiments. The use of local Lipschitz constant is less aggressive on the limited network capacity, which results in better certified robustness.

Clarity:

I believe that the submission needs more spelling checking. Especially in Sec 3.2 which is the main sub-section. I think equation (4) does not make clear sense based on the definitions of those notations.

Significance:

Certified robustness training is an important direction in adversarial robustness researches. Although the proposal is simple, its effectiveness on the improvement can be appreciated by future works.

**Time Spent Reviewing:**

3

---

> ### Author Response · Authors · 2021-08-10
> **Response to reviewer uE4r**
>
> **Q: Notations in Section 3.2 need spelling checking. For instance, Eq 4 does not make sense based on the notations.**
>
> A: We thank the reviewer for carefully reviewing our manuscript. The confusion of (4) might arise from a typo in the definition of $D_V$ in Eq (5). **The correct form of Eq (5) is provided in  Eq (16), Appendix A.2 in the submitted supplementary file.** Specifically,
> D_V should be defined as:
>
> $ D_V^{l}(i, i) = 1(ReLU(z_i^l)>0) $, if  $I_V(i, i) =1 $
>
> $ D_V^{l}(i, i) = 0 $, if  $I_V(i, i)=0 $
>
> where $D_V$ denotes the slope of ReLU outputs for varying neurons. $D_V^{l}(i, i)$ will be 1 (if the ReLU is activated), or will be 0 (if the ReLU is not activated).

---

### Official Review · Reviewer_jfER · 2021-07-13

**Rating:** 5
**Confidence:** 4

**Summary:**

Summary:
The authors propose a training scheme that regularizes against the worst-case logit, as computed by a novel technique for bounding the local Lipschitz constant of a ReLU network.

**Limitations And Societal Impact:**

No limitations were discussed, and negative societal impacts were not thoroughly considered.

**Main Review:**

Originality:
The key novelty in this approach is to use box constraints to determine local information about the stability of ReLU neurons to yield a differentiable upper bound for the L2-Lipschitz constant. Several theoretical results are presented, though it is unclear if the authors' intended them to be taken as novel results: in particular, Theorem and Proposition 1 are known in the literature and can be explained as a simple application of Cauchy's interlacing theorem.  The application of a local Lipschitz upper bound to generate worst-case logits to be used is training is reminiscent of (and should be compared to) other differentiable techniques to generate worst-case logits for use in training, such as DiffAI. Similarly, the hinge-loss for IBP and the notion of regularizing to enforce ReLU stability is similar to the approach in "Training for Faster Adversarial Robustness Verification via Inducing ReLU Stability" by Xiao et al and this should be cited/compared against.

Quality:
The proposed Lipschitz estimation technique is sound, efficient, and differentiable; and while the theoretical underpinnings of removing rows/columns from matrices reducing their spectral radii is known, to the best of my knowledge, it had not been used in such a fashion to generate a Lipschitz upper bound. My primary complaint with this paper is the lack of thoroughness and comparisons in the experimental section. With respect to training a network to have a low Lipschitz constant and/or be adversarially robust, non-certifiable training methods such as AT/TRADES/etc should be considered and compared against (as these are known to regularize the Lipschitz constant of neural networks and provide robustness), as well as the other non-Lipschitz-based certifiable training methods. With respect to the proposed hinge loss for enforcing ReLU stability, the techniques proposed in Xiao et al (see above) should be considered. With respect to this Lipschitz estimate as a certification procedure, comparisons should be made against other techniques designed to certify a fixed, post-training, network (see the VNN challenge).

Clarity:
While the paper was fairly easy to read follow, and explained the key concepts well, this paper could benefit from better organization. Separating and framing this approach as primarily a Lipschitz estimator, that can then be used during training to provide robustness, or alternatively that can be used after training to certify robustness would help clarify the novelties. The ReluTheta activation could then be presented as a means to encourage ReLU stability and thereby tighten the Lipschitz bound, in an orthogonal direction to other such relu-stability techniques.

Significance: Training neural networks that are certifiably robust against adversarial examples is a very important problem that has received significant attention from the community. Provable guarantees leveraging (local) Lipschitz continuity are a promising approach, however it is not clear to me that the proposed approach in this work advances the state of the art in a demonstrable fashion.

Overall:
This approach is novel and promising. However it requires more thorough experimentation and comparison to other Lipschitz--approximation/robustness-certification/relu-stability techniques to be worthy of acceptance, in my opinion.

**** Edited score after reading reviewer rebuttals from 3->5 *********

**Time Spent Reviewing:**

3

---

> ### Author Response · Authors · 2021-08-10
> **Response to reviewer jfER**
>
> **Q: Comparison to Xiao et al.**
>
> **A: We compared our method with the approach proposed by Xiao et al.. Our method achieves better clean, verified and PGD accuracy than theirs.**
> Xiao et al. proposed a Linf norm certified defense by imposing relu stability regularizer with adversarial training, and verifying the network using a mixed integer linear programming (MILP) verifier. The original approach in Xiao et al. is not directly applicable to our setting, as they relied on the MILP verifier which cannot scale to the large models evaluated in our paper, and they focused on Linf norm robustness. To make a fair comparison to their approach, we made a few extensions to their paper:
> 1.  We use L2 norm adversarial training to replace their Linf norm adversarial training;
> 2.  We use the same large CIFAR network (4C3F with 62464 neurons) as in our other experiments;
> 3.  We use the best NN verifier in the very recent VNN COMP 2021 [a], alpha-beta-CROWN [b], to replace their MILP based verifier which is a bit outdated now.
>
> Additionally, we tried different regularization parameters and reported the best results here.
>
> Xiao et al. CIFAR-10 4C3F (verified by alpha-beta-CROWN):
>
> Clean accuray: 57.39%, Verified Accuracy: 51.09%, PGD accuracy: 52.41%
>
> This approach produces a reasonably robust model, thanks to the very recent strong NN verifier. However, its clean, verified and PGD accuracy are worse than ours. Additionally this approach is much less scalable than ours - the verification takes about 150 GPU hours to finish, while our approach takes only 7 minutes to verify the entire dataset. We will cite Xiao et al. and include these results in our final version.
>
> **Q: Comparison to techniques designed to certify a fixed, post-training network (VNN challenge).**
>
> **A: We used the state-of-the-art NN verifier (alpha-beta-CROWN [b], released in July) from the VNN challenge to certify a fixed, post-training network trained via techniques in Xiao et al. and results are presented above.**
> Additionally, we also used alpha-beta-CROWN to certify an purely adversarially trained CIFAR-10 4C3F network. Unfortunately the verified accuracy is close to 0% in this setting.
>
> We agree with the reviewer that employing techniques designed to certify a post-training network is a promising approach, especially SOTA NN verifiers have been greatly improved very recently in VNN COMP 2021. However this is an approach orthogonal to ours and currently no existing works have demonstrated results using the latest NN verifiers, especially in the L2 norm robustness setting. We believe this is a good opportunity for our further work.
>
> **Q: Comparison to other certified defenses such as DiffAI.**
>
> **A: We compared our method to a recent certified defense called CROWN-IBP (based on the same principle as DiffAI), and our method performs about 13% better than the baseline.**
> We thank the reviewer for pointing out other certified defenses generating worst case logits such as DiffAI [c]. Many of these approaches only apply to Linf norm perturbation robustness, and we double checked the DiffAI implementation indeed only supports Linf perturbation and is not directly comparable to ours. However, we find another more recent and stronger certified defense for Linf norm, CROWN-IBP [d], which is based on the same principle as DiffAI (generating worst case logits) and its code also works for L2 norm. We train with CROWN-IBP on MNIST and CIFAR-10 and obtained the following results:
>
> | Dataset | Epsilon | Architecture | Clean (%) | PGD (%) | Certified (%) |
> |---------|---------|--------------|-----------|---------|---------------|
> | MNIST   | 1.58    | 4C3F         | 82.30     | 80.40   | 41.31         |
> | CIFAR   | 36/255  | 4C3F         | 54.20     | 52.70   | 41.90         |
> | CIFAR   | 36/255  | 6C2F         | 53.71     | 52.20   | 41.90         |
>
> The performance gap between these models and our approach is quite big, indicating that our approach is more suitable for L2 norm certified robustness.
> We will cite and discuss the DiffAI paper and add these additional results in our final revision.
>
> [a] VNN COMP 2021 presentation: https://docs.google.com/presentation/d/1oM3NqqU03EUqgQVc3bGK2ENgHa57u-W6Q63Vflkv000/edit
>
> [b] alpha-beta-CROWN: https://github.com/huanzhang12/alpha-beta-CROWN
>
> [c] Mirman, M., Gehr, T., & Vechev, M. . Differentiable abstract interpretation for provably robust neural networks. ICML 2018.
>
> [d] Zhang, H., Chen, H., Xiao, C., Gowal, S., Stanforth, R., Li, B., ... & Hsieh, C. J.. Towards stable and efficient training of verifiably robust neural networks. ICLR 2020.
>
> **Q: Comparison to AT/TRADES (as these are known to regularize the Lipschitz constant of neural networks and provide robustness).**
>
> **A: We compared our method with AT/TRADES methods. Although AT/TRADES regularize the neural network Lipschitz compared to a naturally trained neural network, it is still not enough to provide certified robustness.**
> We agree with the reviewer that AT/TRADES regularizes the Lipchitz constant of neural networks. We train with AT/TRADES on CIFAR-10 with two architectures and report their accuracies, global and local Lipschitz bounds as follows. The certified accuracy is calculated using local Lipschitz bound. As we can see from the table, although the models trained via AT/TRADES have much smaller Lipschitz bound than naturally trained models (we only use cross entropy loss on clean images in natural training), the Lipschitz bound is still too large to give certified robustness.
> As mentioned in the previous question, we also used alpha-beta-CROWN to certify the adversarially trained CIFAR-10 4C3F network. The verified accuracy is still 0%.
>
> | Architecture | method  | Clean | PGD   | Certified | Global Lipschitz | Local Lipschitz |
> |--------------|---------|-------|-------|-----------|------------------|-----------------|
> | 4C3F         | Natural | 80.06   | 59.20     |  0.0     |   14394.97     |   11111.06      |
> | 4C3F         | AT      | 79.62 | 69.9  | 0.0       | 8102.34          | 6637.68         |
> | 4C3F         | TRADES  | 78.95 | 71.47 | 0.0       | 6087.50          | 4855.47         |
> | 6C2F         | Natural |  87.7 | 35.9  | 0.0       | 1520839424.0     | 480923040.0     |
> | 6C2F         | AT      | 80.7  | 70.76 | 0.0       | 22666.5          | 18529.47        |
> | 6C2F         | TRADES  | 80.00 | 72.29 | 0.0       | 18209.72         | 13227.19        |
>
> **Overall,** we have conducted thorough experiments suggested by the reviewer. We hope these new results address your concerns, and we will really appreciate it if you can reevaluate our paper based on these new results, and let us know if you have any additional concerns.

---

> > ### Comment · Reviewer_jfER · 2021-08-19
> > **Response to response**
> >
> > Thank you for making the effort to run additional experiments. I'm generally quite happy with your experiments and willing to raise my score. In particular, the comparisons to CROWN-IBP and more standard adversarial training techniques were very helpful (comparable uncertified robustness to AT/TRADES is surprising!). Indeed, the clarification that verifying post-training networks as an 'orthogonal' line of work was useful, and under this lens and the updated collection of experiments demonstrates the value of this approach.
> >
> > That said, I do think you may have misinterpreted one of my desired experiments:
> >
> > **Xiao et al: Comparing regularizers that influence relu stability.**
> >
> > I'm not exactly sure about the details of this first presented experiment, so let me be clear what I was looking for:
> > I'm interpreting the major crux of your technique as the gains in local Lipschitz estimation that follows from guaranteeing ReLU stability. To this end, a regularizer is proposed to encourage more fixed neurons, with the aim that two things occur: 1) more neurons are fixed, as in figure 3; 2) the estimated Lipschitz constant is tightened. Xiao et al propose multiple methods for inducing ReLU stability, including L1 regularization, weight pruning, and the RS loss (all of which are applicable in your setting, as the intermediate bounds are differentiable functions of the network parameters). I would like to see a comparison, in particular of how incorporating any or all of these techniques affects only the 1) number of fixed ReLus; and 2) how this affects the reported Lipschitz bound. Essentially, if the power in this technique comes from tight local Lipschitz bounds, then does inducing sparsity to yield tighter Lipschitz bounds improve your method?
> >
> > **Overall** after reading the concerns of other reviewers and your rebuttals to same, I now think this paper is borderline.

---

> > > ### Author Response · Authors · 2021-08-26
> > > **Response to reviewer jfER**
> > >
> > > We sincerely thank the reviewer for clarifying the question. We have **conducted new experiments** according to your suggestions. In the meanwhile, we respectfully point out that there seems to be a **misunderstanding on the main contribution** of our paper.
> > >
> > > The regularizer proposed to encourage more fixed neurons is not our main contribution. In fact, the two things - 1) more neurons are fixed, as in figure 3; 2) the estimated Lipschitz constant is tightened - occur even **without** the regularizer to encourage more fixed neurons (experiments in Figure 3 actually do not have the regularizer enabled). **The tightened Lipschitz constant is mainly resulted from computing the local rather than global Lipschitz constant**, and during training the network also learns to make the local Lipschitz constant tighter. As a consequence of using our tighter local Lipschitz bound (**without any regularizer**), we get a certified accuracy gain of around 3%. **Adding a sparsity regularizer only additionally improves the results in a very minor way** (~0.1% for CIFAR, ~0.7% for Tiny-imagenet).
> > >
> > > Therefore, regarding your question: *if the power in this technique comes from tight local Lipschitz bounds, then does inducing sparsity to yield tighter Lipschitz bounds improve your method?*, the answer is: the sparsity loss is only a minor contributor to our performance improvements, and the main contribution of our paper is to efficiently train a certifiably robust model via *local* Lipschiz bounds, which were never explored by previous works.
> > >
> > > As you suggested, we incorporated the three regularizers proposed in Xiao et al. in our training procedure and reported the results in the following table. We report the following metrics:
> > >
> > > 1) *The neurons with fixed values* (e.g., ReLU outputs a fixed 0) correspond to the rows and columns that we can be deleted when calculating a tighter local Lipschitz
> > > 2) *The neurons with fixed signs* are the neurons that ReLU stability loss is trying to promote. Note that the ReLU stability loss has a misaligned goal - it makes the lower bound and upper bound of a neuron have the same sign, and only one of the two cases (both signs are negative) help us to get a tighter local Lipschitz.
> > > 3) We also report local and global Lipschitz constants as you suggested.
> > >
> > > In the table, our approach without any regularizers (54.28%) noticeably improves upon the BCP baseline (51.30%). Then, we report three methods used in Xiao et al.:
> > >
> > > 1) *The ReLU stability regularizer* only very minorly improves model performance (<0.1%), although it does slightly increase the number of neurons with fixed signs. We tried different regularization strengths and reported the non-diverging best results here.
> > > 2) *The L1 regularizer* increases the proportion of neurons with fixed values and with fixed signs by a large amount, however, the global Lipschitz bound is reduced significantly as well, detrimenting the model capacity and accuracy.
> > > 3) *The weight pruning approach* does not seem to be helpful to increase either neuron sparsity or accuracy.
> > >
> > > |                                               | Proportion of Neurons  with fixed values | Proportion of Neurons  with fixed signs | Local Lipschitz bound | Global Lipschitz bound | Clean | PGD   | Certified |
> > > |-----------------------------------------------|------------------------------------------|-----------------------------------------|-----------------------|------------------------|-------|-------|-----------|
> > > | Standard                                      | 0.080                                    | 0.3164                                  | 1520839424.0          | 480923040.0            | 87.7  | 35.9  | 0.0       |
> > > | BCP                                           | 0.5482                                   | 0.7527                                  | 11.08                 | 11.35                  | 65.7  | 60.8  | 51.30     |
> > > | Local-ReLU (ours, no regularization)          | 0.4032                                   | 0.6446                                  | 6.6857                | 7.8903                 | 70.7  | 64.8  | 54.28     |
> > > | Local-ReLU + relu stability loss [Xiao et al] | 0.3995                                   | 0.6487                                  | 6.8538                | 8.1000                 | 70.45 | 64.69 | 54.34     |
> > > | Local-ReLU + L1 regularization [Xiao et al]   | 0.8163                                   | 0.9378                                  | 0.5238                | 0.6108                 | 60.90 | 55.40 | 48.60     |
> > > | Local-ReLU + weight pruning [Xiao et al]      | 0.1076                                   | 0.5786                                  | 4.4672                | 4.9700                 | 65.59 | 60.25 | 49.95     |
> > >
> > > **Overall**, these results indicate that there is a balance between the number of fixed neurons and the model capacity. Training with our local Lipschitz bound achieves a good balance, and adding sparsity regularizers may provide very minor improvements, but that is not the main source of our improvements.
> > >
> > > Finally, we hope the reviewer can also check out our general response, which addressed an issue in a baseline paper (Gloro). The Gloro paper was updated on arxiv **after** NeurIPS deadline, which confused reviewer Miuz so the reviewer gave us a low score. The Gloro paper changed the activation function from ReLU to MaxMin, leading to improved performance. When we also employ the MaxMin activation function, we still consistently outperform their approach thanks to the tighter local Lipschitz.
> > >
> > > We thank the reviewer again for the very constructive comments, and please let us know if you have any further comments or suggestions.

---

> > > ### Author Response · Authors · 2021-09-02
> > > **We hope the reviewer can check out our followup response**
> > >
> > > We thank the reviewer again for the helpful followup comments regarding our response. Since the discussion period is ending soon, we hope the reviewer can take a look at our new response. We respectfully point out that there seems to be a **misunderstanding on the main contribution** of our paper.
> > >
> > > In summary, the sparsity loss is **only a minor contributor** (~0.1% on CIFAR) to our performance improvements and is **not our main contribution**. Our main contribution is to **efficiently train a certifiably robust model via local Lipschiz bounds** which were never explored by previous works. Even without any regularizers we still noticeably outperform baselines.
> > >
> > > As you suggested, we also incorporated the **three regularizers proposed in Xiao et al.** in our training procedure and reported the results in [the table](https://openreview.net/forum?id=FTt28RYj5Pc&noteId=633v30M3zQG). The results indicate that there is a tradeoff among the number of fixed neurons, local Lipschitz constant and the model performance. Introducing the RS regularizers may provide *very minor* improvements, but that is not the main source of our improvements. L1 regularization and weight pruning are not very effective in our setting of L2 norm defense.
> > >
> > > We thank the reviewer again for the constructive comments, and we hope our new response has addressed your remaining concerns. Please kindly let us know if you have further questions.
> > >
> > > Thanks,
> > > Anonymous Authors

---

### Official Review · Reviewer_Miuz · 2021-07-14

**Rating:** 5
**Confidence:** 2

**Summary:**

This paper proposes a new method for certified robustness that tightly bounds the local Lipschitz constants for models by exploiting zero-activation of ReLU. For each data point, the proposed method first calculates the positions where feature maps become zero by ReLU. Next, since the row vector of the weight matrix before the ReLU outputs zero does not affect the output, the proposed method eliminates such row vectors and computes the Lipschitz constants of the modified weight matrix. Experiments demonstrate that the proposed method outperforms previous methods in terms of clean accuracy, robust accuracy against PGD, and certified robust accuracy.


**Limitations And Societal Impact:**

Yes

**Main Review:**

The proposed approach is moderately interesting and its idea might inspire researchers of adversarial robustness. However, I think the technical contributions are not high and experimental results seem to be not very strong. Therefore, I tend to vote to reject.

- Strength
   1. The idea that focuses on zero activations for ReLU is interesting. Ignoring weight vectors corresponding to these zero activations can contribute to more robust models or a better trade-off between clean accuracy and robustness.

   1. Experiments demonstrate the proposed method outperforms other methods. These baseline methods seem to be sufficiently new and reasonable though I do not have very expertise in this research area.

   1. This paper is well written and easy to follow.


- Weakness
  1. Experimental results are not very strong.
In Table 1, the proposed method outperforms other methods. However, accuracies for GloRo in Table 1  are A few percentage points less than those reported in the original paper [4]. In addition, reported accuracies in [4] are higher than those of the proposed method in Table 1. I would like to know why these results are different from the original results.

  1. Theoretical results are trivial, and the technical contribution is not very high. In fact, this paper does not provide proof of Theorem 1 because this result is obvious. Proposition 1 is also a trivial result, which is broadly known results in the matrix computation.
The techniques that constitute the proposed method are not very new. For example, the proposed method naively uses previous methods for computing upper and lower bounds of pre-activations for certifying zero-activation of ReLUs, and the proposed method exploits a power method to compute Lipschitz constants.


- Minor comments
  1. The proposed method is based on the combination of ReLUs and linear operations. Can you use the proposed method with batch normalization? In addition, can you use the proposed method in the ResNet? [15] uses Wide ResNet for evaluation of BCP in , but this paper does not.
  1. Are matrices D in equations (5) and (11) correct? If they are correct, I think the computation of each layer becomes $z=ReLU(Wx)\odot Wx$.

**Time Spent Reviewing:**

6 hours

---

> ### Author Response · Authors · 2021-08-10
> **Response to reviewer Miuz**
>
> **Q: Mismatch between the reported Gloro [1] accuracy (58.4%) and the accuracy in our submission (49.3%).**
>
> A: First of all, thanks for carefully checking the baselines. As mentioned in our general response, the accuracy mismatch is caused by a GloRo paper update **after NeurIPS submission deadline**. Our numbers were based on the [v1 version of GloRo paper](https://arxiv.org/pdf/2102.08452v1.pdf) using the ReLU activation function.
>
> **Q: Theoretical results are trivial. The techniques that constitute the proposed method are not very new.**
>
> **A: Our main contribution is to provide an efficient method in finding a local Lipschitz bound, and incorporate the tighter bound for certifiable training. To the best of our knowledge, this is the first paper incorporating local Lipschitz bound in training, and can have great practical impact.**
> The main purpose of Theorem 1 and Proposition 1 is to demonstrate that the proposed local Lipschitz bound is always provably tighter than the global Lipschitz bound. Although the theory is simple, the majority contribution of our paper is to apply it for the important challenge of training certifiably robust networks, and we achieve SOTA results. Our method can be used as a plug-in module to many existing certifiable training algorithms, and leads to probably better certifiable robustness accuracy by tightening the Lipschitz bound.
>
> **Minor comments**
>
> **Q: Can you use the proposed method with batch normalization?**
>
> **A: Yes, we can use the proposed method with batch normalization.** The Lipschitz constant of batch normalization is (1/std*scale), where std is the batch standard deviation during training, and running_mean during testing; scale is a learnable parameter in batch normalization.
>
> **Q: In addition, can you use the proposed method in the ResNet? [15] uses Wide ResNet for evaluation of BCP in , but this paper does not.**
>
> **A: Yes, we can apply our method in ResNet.**
> The main differences between ResNet and the architectures that we use are that Resnet has skipping connections and batch normalization. Let the input be $x$, the output $y$ of a basic block is $y = F(x) + x = \text{Conv}_2(\text{ReLU} (\text{BN} (\text{Conv}_1 \text{ReLU}( \text{BN} (x) )))) + x$.
> For the batch normalization layer, we compute its Lipschitz bound as answered in the previous question.
> For the ReLU activations, we can apply our method to obtain varying and constant neurons, so we can delete rows and columns in the convolutional weights.
> For the skipping connection, we need to use triangular inequality to bound its Lipschitz, i.e. $ || F + I || \leq || F || + 1 $, where $||F||$ is the local Lipschitz bound.
> In fact, the ResNet architecture is not very suitable for certified robustness compared with smaller conv nets such as 6C2F for several reasons. 1) The skipping connection expands the Lipschitz because we need to use triangular inequality to get the upper bound. 2) The depth of ResNet makes it hard to get a tight Lipschitz bound.
>
> We did notice that [15] also uses Wide ResNet for evaluation, but the Wide ResNet architecture performs worse than the 6C2F architecture in terms of certified robustness. In addition, [15] did not provide either the source code or the detailed description of the Wide ResNet architecture that they use for us to reproduce the results.
>
> **Q: Are matrices D in equations (5) and (11) correct? If they are correct, I think the computation of each layer becomes z=ReLU(Wx)⊙Wx.**
>
> A: We thank the reviewer for pointing out the typo. The correct form of Eq (5) and (11) are provided in Eq (16) in Appendix A.2.
> A: We thank the reviewer for pointing out the typo. As mentioned in the general response, **the correct form of Eq (5) is provided in  Eq (16), Appendix A.2 in the submitted supplementary file.** Specifically,
> D_V should be defined as:
>
> $ D_V^{l}(i, i) = 1(ReLU(z_i^l)>0) $, if  $I_V(i, i) =1 $
>
> $ D_V^{l}(i, i) = 0 $, if  $I_V(i, i)=0 $
>
> where $D_V$ denotes the slope of ReLU outputs for varying neurons. $D_V^{l}(i, i)$ will be 1 (if the ReLU is activated), or will be 0 (if the ReLU is not activated).
>
> **Overall**, we have adapted our method to the updated version of the Gloro baseline and outperform strong SOTA baselines. We also clarified our contribution and discussed the extension of our method to other architectures. We hope the reviewer can reevaluate our paper based on these new results.

---

> ### Author Response · Authors · 2021-08-27
> **We hope the reviewer can check out our response and reevaluate our paper**
>
> We thank the reviewer again for the helpful comments. Since the discussion period is ending soon, we hope the reviewer can take a look at our response, since the main weakness pointed out by the reviewer is mostly due to misunderstandings.
>
> Importantly, our reported accuracies for GloRo are different from their original paper because **their paper was updated on arxiv after NeurIPS deadline**, where they changed the activation function from ReLU (49.3% accuracy) to MaxMin (58.4% accuracy). When we also employ the same MaxMin activation function, we still consistently outperform their approach (we achieve 60.7% accuracy, see details in general response).
>
> We also addressed other technical questions in our response. We hope the reviewer can re-evaluate our paper, and please kindly let us know if you have further questions.
>
> Thanks,
> Anonymous Authors

---

> > ### Comment · Reviewer_Miuz · 2021-08-31
> > **Thank you for your reply.**
> >
> > I have read all reviews and feedback. I think the feedback addresses the issue about experiments very well. I think the proposed method is reasonable, and there are no technical flaws. On the other hand, I think that it is not a very surprising method, and its technical depth is not very high. In fact,  the proposed method requires a large overhead to improve robust accuracy by a few percentage points. Thus, I maintain my score.

---

> > > ### Author Response · Authors · 2021-09-01
> > > **Thank you for reading our feedback! Some comments on our improvements and the overhead.**
> > >
> > > We sincerely thank the reviewer for checking out our response, and we are happy to know that our feedback addressed your issues!
> > >
> > > We would still like to highlight our technical contribution here. **To the best of our knowledge, this is the first paper incorporating local Lipschitz bound in certified training.** Our proposed method can be plugged into any method that uses global Lipschitz bound and provides *provable* improvement. We do agree that computing the local Lipschitz constant itself does not involve in-depth techniques, but we focus on applying it to the certified training procedure and demonstrating its practical advantages. In fact, a successful training method cannot be too complex due to efficiency reasons, and previous works were also based on simple principles such as the global Lipschitz bound.
> > >
> > > As for the amount of improvement, we note that in the certified robustness community a few percentage points is very hard due to the hardness of the problem. For example, BCP (**NeurIPS 2020**) improves the robustness accuracy of the baseline CAP (**NeurIPS 2018**) from **50.87%** to **51.30%**; Gloro (**ICML 2021**) has a certified accuracy of **51.0%** on the same ReLU network. We improve the robust accuracy to **54.3%** under the same setting which is a relatively large improvement compared to improvements in previous works.
> > >
> > > As for the overhead, at training time, our method increases the training time of BCP by ~1.16 times on TinyImageNet and ~2.7 times on CIFAR-10, but considering our improved certified accuracy this overhead is quite reasonable. At test time, to compute robustness certificates, the main overhead is to run power iteration for each input. However, we can exclude samples that can be trivially certified by the global Lipschitz bound, and samples that can be attacked by PGD attack. Eventually, we only need to run power iterations for ~13.8% test samples, so the overhead is also quite reasonable. We discussed this point from line 307 to line 310 in our submission.
> > >
> > > Overall, we do believe that our paper has made an important improvement in the field of certified adversarial defense, and our overhead for both training and test is reasonable. Thank you again for your response and please kindly let us know if you have any additional questions or suggestions.

---

### Official Review · Reviewer_u352 · 2021-07-17

**Rating:** 6
**Confidence:** 4

**Summary:**

This paper provides a trainable local Lipschitz upper-bound for a neural network that is tighter than the global Lipschitz upper-bound computed via multiplying the Lipschitz constant of each individual layer. By making use of the interaction between the weight matrices and the piece-wise linear activation functions, the authors demonstrate that the global Lipschitz bound could be improved when the activation functions are constant locally. In addition, the authors proposed a variant of ReLU that is clipped at a certain upper threshold to increase the constant region to help achieve a tighter Lipschitz upper-bound. Empirically, the authors showed that optimizing the local Lipschitz bound, along with using the modified ReLU, improved the certified robust performance of existing algorithms on several standard image classification tasks.

**Limitations And Societal Impact:**

One major limitation of the work is that it would only tighten the Lipschitz constant bound when the activation functions have constant region (e.g., the output of the activation function is constant with a range of input). This can be an issue nowadays because it precludes the usage of activations without constant regions (e.g., MaxMin/OPLU [3, 5]). In particular, MaxMin has adapted in several recent works in Lipschitz-constrained neural networks and achieved better certified robust accuracy compared against ReLU (e.g., [1, 2, 4]).

[1] Asher Trockman and J Zico Kolter. Orthogonalizing convolutional layers with the cayley transform. In International Conference on Learning Representations, 2021. URL https://openreview.net/forum?id=Pbj8H_jEHYv.

[2] Klas Leino, Zifan Wang, and Matt Fredrikson. Globally-robust neural networks. In Marina Meila and Tong Zhang, editors, Proceedings of the 38th International Conference on Machine Learning, volume 139 of Proceedings of Machine Learning Research, pages 6212–6222. PMLR, 18–24 Jul 2021. URL http://proceedings.mlr.press/v139/leino21a.html.

[3] Cem Anil, James Lucas, and Roger Grosse. Sorting out Lipschitz function approximation. In Kamalika Chaudhuri and Ruslan Salakhutdinov, editors, Proceedings of the 36th International Conference on Machine Learning, volume 97 of Proceedings of Machine Learning Research, pages 291–301. PMLR, 09–15 Jun 2019. URL http://proceedings.mlr.press/v97/anil19a.html.

[4] Qiyang  Li,  Saminul  Haque,  Cem  Anil,  James  Lucas,  Roger  B  Grosse,  and  Joern-Henrik  Jacobsen. Preventing  gradient  attenuation  in  lipschitz  constrained  convolutional  networks. In  H.  Wallach,   H.  Larochelle,   A.  Beygelzimer,   F.  d'Alch ́e-Buc,   E.  Fox,   and  R.  Garnett,   editors, Advances  in  Neural  Information  Processing  Systems,   volume  32.  Curran Associates,   Inc.,   2019. URL https://proceedings.neurips.cc/paper/2019/file/1ce3e6e3f452828e23a0c94572bef9d9-Paper.pdf.

[5] Artem Chernodub and Dimitri Nowicki. Norm-preserving orthogonal permutation linear unit activation functions (oplu). arXiv preprint arXiv:1604.02313, 2016.

**Main Review:**

## Strengths
*Originality* - The proposed approach is elegant and seems to be easy to implement in existing neural network architectures.

*Quality* - Theorem statements all make intuitive sense and are likely to be sound (though I admittedly did not carefully check all the proofs).

*Clarity* - The paper is well-written and easy to follow.

## Weaknesses
*Significance*
1. Several important baselines are missing from the experimental comparisons (e.g., [1, 4]). Also, why is the performance of GloRo reported in the current paper (Table 1, 49.0% certified accuracy for CIFAR-10) much weaker than the performance reported in the original GloRo paper? (Table 1, 58.4% certified accuracy for CIFAR10) [2].
2. The proposed approach seems to successfully improve the performance of ReLU(-like) networks, but the best reported robust accuracy is still worse than existing Lipschitz-constrained models. For example, the best model in the current paper has a certified accuracy of 54.3% on CIFAR-10 benchmark while GloRo has already achieved 58.4% certified accuracy for the same benchmark.

Overall, I think this paper presents an interesting approach that tightens the local Lipschitz bound, but the current empirical evaluations seem to suggest that the proposed approach has a weaker performance than some baselines in the literature. I wouldn't recommend acceptance before more thorough experiments and comparisons are made with these baselines.

(See the limitation section for the references)

***** Edit *****
Based on the new results on MaxMin activation functions, I have decided to raise my score to 6.

**Time Spent Reviewing:**

5

---

> ### Author Response · Authors · 2021-08-10
> **Response to reviewer u352**
>
> **Q: Mismatch between the reported Gloro accuracy (58.4%) and the accuracy in our submission (49.3%).**
> First of all, thanks for carefully checking the baselines. As mentioned in our general response, the accuracy mismatch is caused by a GloRo paper update **after NeurIPS submission deadline**. Our numbers were based on the [v1 version of GloRo paper](https://arxiv.org/pdf/2102.08452v1.pdf) using the ReLU activation function.
>
> **Q: Comparisons with the latest baselines with non-ReLU activation functions.**
>
> A: We sincerely thank the reviewer for pointing to us a lot of related papers using non-ReLU activation functions.
> Our idea of exploiting local information is general and can be applied to a modified MaxMin function, and **our method outperforms two strong baselines [1,2] with 60.7% certified accuracy on CIFAR-10**. For the details, please refer to the general response for details.
> We totally agree with the reviewer that the development of non-ReLU activation functions is interesting to the robustness community. But we believe the main insight of this paper is to leverage local information in order to achieve a tighter Lipschitz bound, and this technique can be adapted to new activation functions.
> We will add a subsection in our paper discussing non-ReLU activations and our general approach to handle these situations, and include new results based on the MaxMin activation function.
>
> **Overall**, we have demonstrated that our approach can be applied to the MaxMin activation function and we can also outperform strong SOTA baselines. We hope the reviewer can reevaluate our paper based on these new results.

---

### Official Review · Reviewer_umTB · 2021-07-23

**Rating:** 7
**Confidence:** 3

**Summary:**

Per data-example, a local Lipschitz bound on the neural network output is computed and then used to certify robustness. This bound is made tighter by exploiting invariances of the ReLU activations throughout the network under small perturbations around the given data point. The paper further increases invariances by clipping ReLU and training the clipping threshold.

**Limitations And Societal Impact:**

No immediate societal impact.

**Main Review:**

Strengths:
+ Simple and computationally efficient method of obtaining tight local Lipschitz bound makes this algorithm practical for usage in training
+ The Lipschitz bound improvement relies only on ReLU outputs being constant in some region. This makes the method plug-and-play and also applicable to most modern neural networks.
+ Experimental results show good improvements over baselines in terms of the bound, training loss, and final accuracy

Weaknesses / suggestions:
- Computational cost is explored in terms of number of iterations of the power method. A more appropriate metric would be total number of floating point operations or compute time. A comparison along these metrics can then be made with the other methods that use global Lipschitz bounds (such as GloRo)
- The clipped version of ReLU can make it harder for the network to learn since there are more regions with zero-gradients. Does this happen? Are there ways this was overcome (e.g. through reducing the magnitude of the initialized weights?). This could be explored further to explore the strengths and limitations of the ReLU_theta approach
- For local Lipschitz bound, the authors split the expression for F(x, W) in Eqn. 4 into a product of matrices, and then take the spectral norm of each. However, a direct spectral norm of the entire matrix that premultiplies x can be considered. This would provide an even tighter local Lipschitz bound due to the sub-multiplicative nature of spectral norm. While this method is likely to incur a large computational penalty (due to matrix-matrix multiplications), it could be tried on smaller problems (e.g. MNIST) and it would be interesting to know how much better the local Lipschitz bound can become and its eventual effect on certified accuracy. An alternative is to consider a different grouping of the matrices.

Typos:
- Abstract: “training algorithms can certified robustness” (grammar)
- Line 147: I_c and I_v definition swapped?
- Eqn 4 should have I_v instead of D_v. (since Wx already equals z_i, and so multiplying with D_v would produce z_i^2, which should not be there.
- Eqn. 10 also seems to have a similar error as above

**Time Spent Reviewing:**

5

---

> ### Author Response · Authors · 2021-08-10
> **Response to reviewer umTB**
>
> **Q: A more appropriate metric for computational cost is the floating point operations or compute time.**
>
> A: Thanks to the helpful suggestion. We agree with the reviewer and we indeed compared the computational time (seconds per training epoch) of our method, CAP, Gloro, and BCP in the supplementary Appendix C.2 Table 6. On MNIST and CIFAR-10 datasets, our cost is 2 to 3 times higher than BCP, because we utilize local information to tighten the Lipschitz constant and outperform these baselines in verified accuracy.
>
> | Data          | Model | CAP  | Gloro | BCP   | Ours  |
> |---------------|-------|------|-------|-------|-------|
> | MNIST         | 4C3F  | 689  | 9.0   | 17.3  | 45.5  |
> | CIFAR-10      | 4C3F  | 645  | \na   | 23.5  | 38.2  |
> | CIFAR-10      | 6C2F  | 1369 | 6.5   | 26.0  | 69.8  |
> | Tiny-Imagenet | 8C2F  | \na  | \na   | 343.5 | 398.8 |
>
> **Q: Zero-gradients issue for ReLU_theta.**
>
> A: Thanks for the interesting question. We thought about this before as well but we didn’t observe any significant problems in training. Another empirical success for the clipped version of ReLU is ReLU-6 (similar to our relu_theta but with a fixed threshold of 6). The widely used MobileNet-v2 [1] uses ReLU-6 and it performs well. Despite the empirical success, we do agree that exploring the gradient issues of activation functions with constant regions is an interesting direction and can be explored in more depth in our future work.
>
> [1] Sandler, M., Howard, A., Zhu, M., Zhmoginov, A., & Chen, L. C. (2018). Mobilenetv2: Inverted residuals and linear bottlenecks. In Proceedings of the IEEE conference on computer vision and pattern recognition (pp. 4510-4520).
>
> **Q: A direct spectral norm of the entire matrix that premultiplies x can be considered.**
>
> A: We appreciate the reviewer’s insightful suggestion on directly bound the Lipschitz constant. Unfortunately, as mentioned in the general response, **computing this bound has exponentially complexity and is infeasible in practice.**  The major challenge for directly computing the spectral norm lies in the $D_V$ terms (i.e., $D_V^{L-1}, D_V^{L-2}, …, D_V^{1}$). Importantly, since $D_V$ denotes the slope of ReLU for the varying neurons, the value of D_V is not determined until the realization of local perturbations. In order to obtain a uniform upper bound, one way is to brute force compute $||(W^L I_V^{L-1}) D_V^{L-1} (I_V^{L-1} W^{L-1} I_V^{L-2} .... D_V^{1} (I_V^{1} W^{1}) x||$ for all possible realizations of $D_V^{L}$, which has $2^n$ complexity and $n$ is the feature map dimension at layer L.
>
> **Q: An alternative is to consider a different grouping of the matrices.**
>
> **A: We discussed an alternative way of grouping matrices in Appendix A.2.**
> Besides directly bounding the spectral norm of the entire matrix that premultiplies x, we indeed thought about an alternative grouping of the matrices, where we decompose the regions of ReLU into zero regions and linear regions, so we can merge all the linear regions to get a tight Lipschitz bound across layers. However, the nonlinear regions will lead to multiple non-negative terms in the final Lipschitz bound due to the triangular inequality. So this alternative approach is not provably tighter than the global Lipschitz bound. In comparison, the proposed method always guarantees tighter Lipschitz bound.
>
> **Q: Typos**
>
> A: We thank the reviewer for carefully reading our paper and pointing out the typos. Indeed, $I_C$ and $I_V$ should be swapped in line 147. The confusion of Eq 4 and Eq 10 is caused by a typo in the definition of $D_V$ (Eq 5). We provided the correct definition of $D_V$ in the general response and in Eq 16 in Appendix A.2.
>
> **Overall**, we have compared the computational cost in terms of computational time. We also discussed alternative ways of grouping matrices for bounding Lipschitz constant. We thank the reviewer for helping improve the paper. If you like our paper, please champion it.

---

### Author Response · Authors · 2021-08-10
**General response**

We thank the reviewers for the detailed comments. We notice there are some common concerns, especially, there were misunderstandings on the results of a baseline (Gloro). We respond to these questions in the general response below.

**Regarding the experiments:**

**Q: Mismatch between the reported Gloro [1] accuracy (58.4%) and the accuracy in our submission (49.3%).**

A: The Gloro paper was **changed on June 11, after NeurIPS submission deadline**, and the numbers (49.3%) reported by us are from their [v1 version on arxiv](https://arxiv.org/pdf/2102.08452v1.pdf) (Table 3(a), CIFAR 6C2F, GloRo). Also, we were also able to reproduce these results on **ReLU** networks. Their new results (58.4%, reported on June 11) were based on the **MaxMin** activation function. Our approach can also be adopted to the MaxMin activation function, obtaining **60.7%** verified accuracy, outperforming theirs (see details below).

**Q: Comparison with the latest baselines with non-ReLU activation functions.**

**A: Our approach can be extended to MaxMin activation functions, and outperforms Gloro [1] and Caylay transform [2] baselines. We incorporated the new baselines into Table 1 of our paper [link](https://paper8401.github.io/local_lipschitz/).**
The main contributor to Gloro’s performance leap is that they change the ReLU activation to MaxMin [3] activation function, which leads to around 10% VRA performance improvement on all MNIST, CIFAR 10 and Tiny-Imagenet experiments.
Our paper proposes the general idea of exploiting local Lipschitz for certified defense, and it can be extended to the MaxMin activation function. Let $x_1$ and $x_2$ be two groups of the input, the output of MaxMin is $\max(x_1, x_2), \min(x_1, x_2)$. To exploit local Lipschitz, we created a clamped version of MaxMin, similar to ReLU_theta. The output of the clamped MaxMin is $\min(\max(x_1, x_2), a), \max(\min(x_1, x_2), b)$, where $a$ is an upper threshold for the max output in MaxMin and $b$ is a lower threshold for the min output in MaxMin. Box propagation rule through MaxMin is straightforward to derive, so we can get the box bound on each entry after MaxMin. If the lower bounds of the Max entries are bigger than the upper threshold, or the upper bounds of the Min entries are smaller than the lower threshold, we can delete the corresponding columns in the successive matrix (similar to the procedure for ReLU networks).

We present our results using MaxMin activation functions below. We consistently outperform in all metrics (clean, PGD and certified accuracy). The improvement is not surprising since none of the existing approaches utilize local information.

|                       | Clean (%) | PGD (%) | Certified (%) |
|-----------------------|-----------|---------|---------------|
| Gloro + MaxMin [1]        | 77.0      | 69.2    | 58.4          |
| Caylay + MaxMin [2]      | 75.3      | 67.7    | 59.2          |
| Ours + Clamped Maxmin | **77.4**      |**70.4**    |**60.7**          |

[1] Klas Leino, Zifan Wang, and Matt Fredrikson. Globally-robust neural networks. In International Conference on Machine Learning, 2021.

[2] Asher Trockman and J Zico Kolter. Orthogonalizing convolutional layers with the cayley transform. In International Conference on Learning Representations, 2021.

**Regarding the method:**

**Q: A direct spectral norm of the entire matrix that premultiplies x can be considered.**

**A: Unfortunately, computing this bound has exponentially complexity and is infeasible in practice.**  Note that directly bounding  the Lipschitz constant by computing $||(W^L I_V^{L-1}) D_V^{L-1} (I_V^{L-1} W^{L-1} I_V^{L-2} .... D_V^{1} (I_V^{1} W^{1}) x||$. Indeed, if one can directly compute the ​​spectral norm of the matrix products, it would be the exact local Lipschitz constant. However, the major challenge for directly computing the spectral norm lies in the $D_V$ terms (i.e., $D_V^{L-1}, D_V^{L-2}, …, D_V^{1}$). Importantly, since $D_V$ denotes the slope of ReLU for the varying neurons, the value of $D_V$ is not determined until the realization of local perturbations. In order to obtain a uniform upper bound, one way is to brute force compute $||(W^L I_V^{L-1}) D_V^{L-1} (I_V^{L-1} W^{L-1} I_V^{L-2} .... D_V^{1} (I_V^{1} W^{1}) x||$ for all possible realizations of $D_V^{L}$, which has $2^n$ complexity and $n$ is the feature map dimension at layer L. For example, consider a 6C2F network (6 conv layers and 2 fully-connected layers), $D_V^{1}$ already contains $2^{32768}$ scenarios, in total, there are $2^{115200}$ scenarios, and it is infeasible to use such an exhaustive method.

**Q: Typo in Eq 5.**

A: We thank the reviewers for carefully reading our paper and pointing out the typo. **The correct definition of $D_V$ is provided in  Eq (16), Appendix A.2 in the submitted supplementary file.**
$D_V$ should be defined as:

$ D_V^{l}(i, i) = 1(ReLU(z_i^l)>0) $, if  $I_V(i, i) =1 $

$ D_V^{l}(i, i) = 0 $, if  $I_V(i, i)=0 $

where $D_V$ denotes the slope of ReLU outputs for varying neurons. $D_V^{l}(i, i)$ will be 1 (if the ReLU is activated), or will be 0 (if the ReLU is not activated).

---

### Decision · Program_Chairs · 2021-09-27

**Decision:**

Accept (Poster)

**Comment:**

All the reviewers agreed that the paper provides interesting results in bounding the (local) Lipschitz constant of NNs with piece-wise linear activation functions. The reviewers had a number of concerns which were mostly resolved after the authors' responses. I would strongly recommend that the authors incorporate all the comments mentioned in the reviews in their updated version (e.g. the additional experimental results, comparison with prior work, etc).